# BEST: Benchmarking Efficiency in Space and Time for LLM-Generated Code

**Aocheng Shen** [* 1 2 3]  **Boyu Zhang** [* 1]  **Jiaze Li** [* 1]  **Ruixuan Ma** [* 1]  **Qiankun Zhang** [1 2 3]  **Jing Wang** [4]
**Bin Yuan** [1 3 5 6 7]  **Shenghao Liu** [1]  **Xianjun Deng** [1 3]

## Abstract

Large language models (LLMs) have revolutionized research in software engineering, and among various tasks, LLM-based code synthesis is promising. A recent line of benchmarks aims to evaluate LLM-generated codes in time efficiency, beyond their correctness. However, *space*, another vital aspect of code efficiency, is rarely evaluated in prior benchmarks. To fill in the gap, this paper introduces *BEST*, the first benchmark for evaluating the efficiency of LLM-generated codes in *both time and space*. It comprises $440$ coding tasks that are rigorously constructed by experts. In addition, we propose a fine-grained *subtask-based* evaluation scheme by dividing each task into multiple subtasks, with different input scales and difficulties. Each subtask is then accompanied by an expert-crafted standard implementation as the efficiency baseline, which achieves the *Pareto optimum*. Building on BEST, we introduce a unified and novel dual-indicator (time and space) metric, named dual@$k$, generalizing the notion of the standard pass@$k$ metric and building on a careful and novel construction of a *weight matrix* of subtasks. Through extensive experiments with dual@$k$ across 50 LLMs on BEST, our evaluation demonstrates that while LLMs exhibit weak capabilities in generating time-efficient code, their capabilities in space-efficient code generation are even worse. The benchmark is provided at https://github.com/kmsgk0/BEST.

---

[*]Equal contribution  [1]School of Cyber Science and Engineering, Huazhong University of Science and Technology, Wuhan, China [2]Key Laboratory of Cyberspace Security, Ministry of Education, Zhengzhou, China [3]Hubei Key Laboratory of Distributed System Security, Wuhan, China [4]School of Software Engineering, Huazhong University of Science and Technology, Wuhan, China [5]Songshan Laboratory, Zhengzhou, China [6]Jinyinhu Laboratory, Wuhan, China [7]Visiting researcher with the Lion Rock Labs of Cyberspace Security, CTIHE, Hong Kong, China. Correspondence to: Qiankun Zhang <qiankun@hust.edu.cn>.

*Proceedings of the 43rd International Conference on Machine Learning*, Seoul, South Korea. PMLR 306, 2026. Copyright 2026 by the author(s).

*You should choose algorithms that use the resources of time and space efficiently.*

*Introduction to Algorithms*, Cormen et al.

## 1. Introduction

The rapid advancement of large language models (LLMs), such as GPT-5.2 (OpenAI, 2025) and Gemini-3 (Google, 2025), has brought significant changes to various tasks in software engineering, among which code generation via LLMs (Chen et al., 2021; Li et al., 2022; Nijkamp et al., 2022; Rozière et al., 2023; Bai et al., 2023; Lozhkov et al., 2024; DeepSeek-AI et al., 2024; Guo et al., 2024) has been verified to be promising and practical. Various benchmarks, such as HumanEval (Chen et al., 2021), MBPP (Austin et al., 2021), APPS (Hendrycks et al., 2021), and DS-1000 (Lai et al., 2022), have been established to evaluate the correctness of LLM-generated codes. Very recently, there has been another series of works, including EffiBench (Huang et al., 2024), Mercury (Du et al., 2024), ENAMEL (Qiu et al., 2024), EvalPerf (Liu et al., 2024b), COFFE (Peng et al., 2025), and EffiBench-X (Qing et al., 2025), aiming at benchmarking the *efficiency in time* of codes for consideration of resource-constrained execution environments. Their works may construct a set of coding tasks as the benchmark, test the running time of codes, and compute various metrics to assess the time efficiency, which provides remarkable insights into understanding the capabilities of LLMs in efficient code generation.

This paper broadens their research scopes based on a textbook-level fact: *code efficiency refers to the optimization of a program's performance by minimizing both time and space needed for execution.* Efficiency in *space*, while comparably important as time efficiency especially in memory-constrained environments such as embedded systems and Internet of Things (IoT) devices, has been overlooked in existing evaluations. Further, another widely-known fact is that algorithm optimization often suffers a *time-space trade-off*, meaning that algorithms trade increased time for decreased space, and vice-versa. Consider an example task presented in Figure 1. In a range sum query task, three algorithms solving it can be various in time and space com-

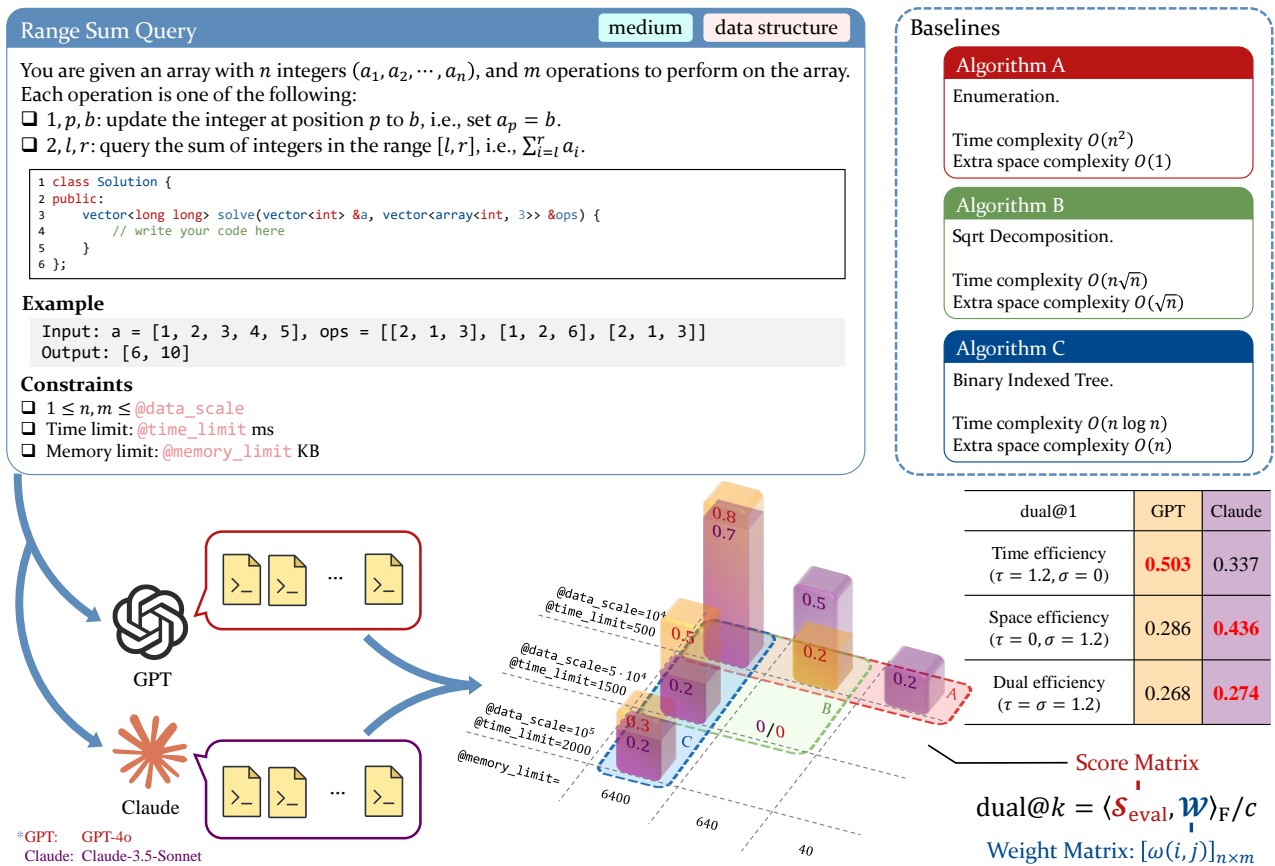

*Figure 1.* An overview of the BEST benchmark and the subtask-based evaluation. An example task in BEST contains 9 subtasks and 3 Pareto optimal baselines. Two LLMs, GPT and Claude, generate codes for each subtask of different ranks in input scales and difficulty and obtain a score matrix. The dual@1 metric is computed based on the score matrix and a thoughtfully designed weight matrix of subtasks. The dual@1 can be easily modified to evaluate time, space, or both efficiency. See Appendix B for details of the subtasks and baselines.

plexity. While the time complexity drops from $O(n^2)$ to $O(n \log n)$, the (extra) space complexity rises from $O(1)$ to $O(n)$. More examples include dynamic programming (DP), hash tables, and so forth. Given such findings, the following limitations and challenges in existing benchmarks need to be addressed:

- **Lack of space efficiency evaluation.** Existing efficiency benchmarks mostly focus on time while overlooking space (Du et al., 2024; Qiu et al., 2024; Liu et al., 2024b; Peng et al., 2025). These benchmarks are deficient in their ability to distinguish between codes of different space complexity as illustrated in Figure 1. EffiBench (Huang et al., 2024) and Effibench-X (Qing et al., 2025) evaluate both time and space efficiency separately in their evaluation. However, their tasks, while abundant, need a careful characterization for a more targeted evaluation.

- **Task selections.** It is easy to collect coding tasks on various online resources, such as Leetcode (LeetCode, 2024). However, only a small fraction of them are suited

for efficiency evaluation. Such tasks are expected to be solved correctly by *multiple* algorithms of *different* time or space complexity. Existing works may roughly collect online coding tasks without a careful selection (Du et al., 2024; Huang et al., 2024; Peng et al., 2025).

- **Task Diversity.** Most recent benchmarks (Austin et al., 2021; Chen et al., 2021; Liu et al., 2023a) contain only elementary coding tasks, failing to substantially challenge LLMs and truly reflect their potential. Other benchmarks (Qiu et al., 2024; Du et al., 2024; Liu et al., 2024b; Peng et al., 2025; Qing et al., 2025), while containing tasks of varying difficulty, do not clearly categorize algorithm types and lack fine-grained evaluation of LLMs.

**Contribution #1: a new benchmark—BEST** Our first contribution is a new benchmark, BEST, to evaluate the efficiency in both time and space for LLM-generated codes. This paper introduces the first version of BEST, which comprises 440 coding tasks, either attentively collected from various online resources or originally created by our experts. All tasks are categorized delicately according to the

algorithms applied and are labeled by their difficulties. To mitigate the risk of data leakage, a new version will be released at regular intervals.

A natural next question is *how to measure the time and space efficiency on BEST*. Our work considers dual efficiency indicators, namely time and space, and there is always a large correlation between them, which poses the following new challenges that may not arise in previous works:

- **Test cases.** In order to distinguish the discrepancies in efficiency between different algorithms, the test cases must be prudently constructed rather than randomly generated (Chen et al., 2021; Austin et al., 2021). In evaluating efficiency, better test cases are expected to accurately distinguish between algorithms of different time and space complexity.

- **Baselines.** Efficiency evaluation proceeds by comparing the performance of the LLM-generated code with a standard implementation as a baseline. Previous works refer to standard codes posted online (Huang et al., 2024; Du et al., 2024; Niu et al., 2024; Peng et al., 2025; Qing et al., 2025), the expert-crafted codes (Qiu et al., 2024), or LLM-generated codes (Liu et al., 2024b). In our task, however, considering a single baseline may not be sufficient. For example, in Figure 1, it is hard to judge which of the three algorithms is optimal or if all three of them are, to some extent, optimal. It is, therefore, not appropriate to apply those time efficiency baselines to evaluate space efficiency directly.

- **Metrics.** The challenge in metrics might be obvious because all metrics defined in previous works, such as Beyond (Du et al., 2024), eff@$k$ (Qiu et al., 2024), DPS (Liu et al., 2024b), or efficient@$k$ (Peng et al., 2025), measure time efficiency only. It is non-trivial to generalize it to a dual-indicator metric. Further, their metrics may be limited by high hardware dependency and inaccurate assessment of the asymptotic performance.

**Contribution #2: a fine-grained subtask-based evaluation scheme.** For a fine-grained evaluation, inspired by the grading policy for coding tasks in the International Olympiad in Informatics [1] (IOI), every task in BEST corresponds to a bunch of *subtasks* with different input scales and time-space limits. For each subtask, a human expert is asked to design multiple test cases within the same rank of the scale and difficulty, as well as a baseline that achieves a *Pareto optimum* [2] (Mascolini, 2015) in efficiency. Separate

---

[1] IOI website: https://ioinformatics.org/

[2] Intuitively, borrowing an idea from the multi-objective optimization, we define a code achieves the Pareto optimum in efficiency when the code cannot simultaneously optimize time and space complexity further. For example, all three algorithms in Figure 1 are Pareto optimal.

and independent evaluation on each of these subtasks forms a score matrix based on the standard pass@$k$ metric (Chen et al., 2021).

**Contribution #3: a novel metric—dual@$k$.** We propose a unified and novel metric, named dual@$k$, to evaluate the code efficiency in both time and space. Given the aforementioned score matrix, we define an additional weight matrix to capture the difficulties among those subtasks. dual@$k$ is basically computed by a Frobenius inner product of these two matrices, and normalized by comparing with the performance achieved by baselines. We thoughtfully design the weight matrix such that dual@$k$: (1) accurately distinguishes between the different capabilities of LLMs in generating efficient codes; (2) is easily adapted to any scenarios where time and space are valued differently.

**Contribution #4: a comprehensive evaluation.** We conduct a comprehensive study to evaluate the efficiency of codes generated by 50 widely-studied LLMs on our BEST. Extensive experiments suggest that while LLMs may be weak at generating time-efficient code, their ability to generate space-efficient code is even weaker. In our evaluation, DeepSeek-V3.2-Thinking and Gemini-3-Pro-Preview excel in both time and space efficiency among open-source and closed-source LLMs, respectively.

## 2. Related Works

Section 1 has discussed in detail the recent works (Qiu et al., 2024; Du et al., 2024; Huang et al., 2024; Liu et al., 2024b; Peng et al., 2025; Qing et al., 2025) that are the most relevant to us in evaluating the efficiency of the LLM-generated code. Table 1 provides a comprehensive comparison to our BEST. We address their limitations regarding datasets, baselines, test cases, metrics, and evaluation schemes in time-space efficiency evaluation. More comparisons will be included in the following sections. Due to the page limitation, other related works are referred to Appendix D.

## 3. BEST Construction

**Overview.** BEST comprises 440 high-quality coding *tasks*. Figure 1 presents an example task. Each task, denoted as $T$, contains a *task statement*, an *interface declaration*, and an *example* of input and output. Inspired by the grading policy of the International Olympiad in Informatics (IOI), BEST adopts a fine-grained *subtask*-based evaluation scheme. Each task comprises multiple subtasks based on different input scales and the corresponding required time and space limits. Each subtask is then accompanied by an

*Table 1.* A comprehensive comparison to existing efficiency benchmarks, including EffiBench (Huang et al., 2024), Mercury (Du et al., 2024), ENAMEL (Qiu et al., 2024), EvalPerf (Liu et al., 2024b), COFFE (Peng et al., 2025), and EffiBench-X (Qing et al., 2025).

| Benchmark | No. of Tasks | Source | Evaluation | Test Case Generation | Baseline | Categorization | Difficulty | Dual Efficiency |
|---|---|---|---|---|---|---|---|---|
| EFFIBENCH | 1000 | LeetCode | Test cases | By LLM | Single std | ✓ | ✓ | ✗ |
| Mercury | 1889 | LeetCode | Test cases | By LLM | Multiple std's | ✗ | ✓ | ✗ |
| ENAMEL | 142 | HumanEval | Level-based | By expert | Single std | ✗ | ✗ | ✗ |
| EvalPerf | 121 | HumanEval+/MBPP+ | Test cases | By LLM | Multiple std's | ✗ | ✗ | ✗ |
| COFFE (Function) | 398 | Various sources | Test cases | By LLM | Single std | ✗ | ✗ | ✗ |
| COFFE (File) | 358 | Various sources | Test cases | By LLM | Multiple std's | ✗ | ✗ | ✗ |
| EFFIBENCH-X | 623 | Various sources | Test cases | By LLM | Single std | ✗ | ✓ | ✗ |
| **BEST (Ours)** | 440 | Various sources | Subtask-based | By expert | Multiple Pareto optimal std's | ✓ | ✓ | ✓ |

expert-crafted [3] standard implementation as the *baseline*. Note that a task may correspond to multiple baselines that achieve a Pareto optimum. Moreover, we *label and categorize* tasks by their difficulty and types of algorithms involved in baselines. Finally, we provide a set of additional *scripts for code judgment*. Table 4 in Appendix A presents a task card that briefly describes each field, which we will discuss in detail below.

**Task collection.** Existing benchmarks (Du et al., 2024; Qiu et al., 2024; Huang et al., 2024) collect tasks from a single source such as LeetCode (LeetCode, 2024) or HumanEval (Chen et al., 2021). In contrast, BEST attentively collects tasks from various online coding contest platforms, such as LeetCode, Codeforces (Codeforces, 2024), AtCoder (AtCoder, 2024), IOI, etc. We also add some tasks that are originally constructed by experts. Further, for consideration of rigorous and comprehensive evaluation, we adopt a *two-stage task selection process*:

- **Implementation-oriented selection.** We expect coding tasks to have varying implementations under different time and space limits. Therefore, we exclude those that have only one implementation, allowing for a more accurate evaluation of LLMs' ability to generate code with varying levels of efficiency.

- **Evaluation-oriented selection.** We further filter the tasks by evaluating them on three LLMs, including GPT-5.2, Qwen3-235B-A22B, and DeepSeek-V3.2, on these tasks. We only retain those tasks for which at least two out of the three LLMs generate different codes in efficiency.

---

[3] We have sufficient (at least five) experts to ensure accuracy and completeness, who has (1) a strong background in algorithms and (2) over 7 years of C++/Python programming experience on average. We also follow a structured, multistage process to construct and verify the baselines and test cases, as detailed in Appendix C.

**Subtask and test cases.** Previous benchmarks either build on the existing HumanEval test cases (Niu et al., 2024), which are recognized as insufficiently challenging (Liu et al., 2023a), or generate test cases through LLMs (Huang et al., 2024), which often produces random instances and may therefore fail to distinguish between suboptimal algorithms. To address it, Qiu et al. (2024) proposes a level-based evaluation, focusing purely on time efficiency. We generalize it by constructing fine-grained subtasks, each of which corresponds to multiple test cases. Specifically, consider a specific coding *task*, denoted as $\mathcal{T}$. We prudently construct $n \times m$ *subtasks* of $\mathcal{T}$, denoted as a subtask matrix $\mathcal{T}_{nm}$, by defining $n$ ranks for time limits and $m$ ranks for space limits (as shown in Figure 1). Each row $\mathcal{T}_{:j}, j \in [m]$ of $\mathcal{T}_{nm}$ shares the same input scale and time limit, while each column $\mathcal{T}_{i:}, i \in [n]$ shares the same space limit. We expect subtasks with larger indices will face stricter time and space limits, indicating that $\mathcal{T}_{ij}$ becomes increasingly difficult with larger values of $i$ or $j$. To achieve this, we ensure that all time and space limits regarding different ranks are carefully designed, so that only codes with the required time and space complexity or better can pass each of them. Also, see the example in Figure 1; Algorithm A can only pass all subtasks of the first column, while Algorithm C can only pass the first row. Finally, each subtask contains multiple test cases that share the same rank in both time and space to ensure a rigorous evaluation of correctness and efficiency.

**Baselines.** Existing benchmarks (Niu et al., 2024; Du et al., 2024; Huang et al., 2024) use standard implementations crafted from online platforms as baselines for evaluation. Qiu et al. (2024) suggest employing human experts to write baseline codes because algorithms and their implementations are often highly non-trivial. We follow their schemes by using expert-crafted baselines. However, recall that in Figure 1, *baselines are not unique* because there always exists a time-space trade-off, meaning that algorithms trade increased time for decreased space, and vice-versa. It leads to an insufficiency to adopt a unique baseline for each

task. Instead, in our BEST, we adopt a specific baseline for each *subtask*. Inspired by the notion of multi-objective optimization, we define a set of *Pareto Optimal* baselines in Definition 3.1.

**Definition 3.1.** A code (or an algorithm) is Pareto optimal in terms of efficiency when it cannot make time efficiency better off without making space efficiency worse off, and vice versa.

For example, in Figure 1, all three algorithms are Pareto optimal by definition. For all tasks in BEST, experts thoughtfully construct Pareto optimal baselines. We remark that the Pareto optimal baseline can be unique for some tasks whose baseline achieves optimal efficiency in both time and space.

**Task labels.** We categorize and label any tasks by their difficulty ('easy', 'medium', and 'hard') and types of algorithms involved in baselines, according to a standard categorization in the algorithm textbook. See Table 5 in Appendix A for the statistics under different labels.

## 4. Metric: dual@$k$

To evaluate the time and space efficiency of an LLM-generated code, a natural approach is to measure the running time and extra memory usage compared to the baseline (Niu et al., 2024; Huang et al., 2024; Du et al., 2024; Qiu et al., 2024). However, such a method has potential drawbacks in: (1) *hardware dependency*. Both running time and memory usage highly depend on execution devices and environments. An ideal metric should remain unaffected by different environments by not directly incorporating runtime and memory usage into its calculation. (2) *inaccurate assessment of the asymptotic performance*. Metrics that consider running time and memory usage do not effectively differentiate code efficiency in small test cases; here, algorithms that are asymptotically inefficient can still yield good performance. Only large-scale test cases can effectively differentiate algorithms with superior asymptotic performance. (3) *dual efficiency measurement*. Previous metrics are typically measured against a single baseline (often the most time-efficient implementation), which fails to provide a comprehensive evaluation of dual optimization capability. (4) *stochasticity of LLM outputs*. The output of LLMs is inherently stochastic, and it is crucial to evaluate the capability of the LLM itself, rather than solely assessing the efficiency of sampled code (Chen et al., 2021).

To address these challenges, first recall that our subtask-based scheme makes only code that satisfies the time or space complexity limits can pass the corresponding subtask, which allows for the effective differentiation of codes with varying asymptotic performance. Additionally, for each subtask $T_{ij}$, rather than evaluating how significantly the actual running time and memory usage surpass the baseline,

we quantify a *score* for each subtask by using the probability that a code passes that subtask (both correctly and efficiently). The pass@$k$ metric (Chen et al., 2021) provides an unbiased estimator on such probability. Thus, we define a *score matrix*, denoted as $\mathcal{S}_{n \times m}$, to record the pass@$k$ on every subtask. Moreover, to capture the varying difficulties of subtasks, we additionally define a *weight matrix*, denoted as $\mathcal{W}_{n \times m}$, which we will explain shortly. Finally, we use the Frobenius inner product of $\mathcal{S}$ and $\mathcal{W}$ to represent the overall score achieved on the task. Our new metric *dual@$k$* is defined as the *ratio* of the overall scores obtained between the *LLM-generated codes* and the *expert-crafted baselines*. Formally, dual@$k$ is defined as:

$$\text{dual}@k \triangleq \frac{\langle \mathcal{S}_{\text{eval}}, \mathcal{W} \rangle_{\text{F}}}{\langle \mathcal{S}_{\text{std}}, \mathcal{W} \rangle_{\text{F}}} = \frac{\mathbf{Tr}(\mathcal{S}_{\text{eval}}^{\mathsf{T}}, \mathcal{W})}{\mathbf{Tr}(\mathcal{S}_{\text{std}}^{\mathsf{T}}, \mathcal{W})} , \qquad (1)$$

where $\mathcal{S}_{\text{eval}}$ and $S_{\text{std}}$ denote the score matrices of the evaluated code and expert-crafted baselines, respectively. In practice, Algorithm 1 gives an implementation to calculate the value of dual@$k$.

**Remark on dual@$k$.** We highlight that the dual@$k$ defined in Eqn. (1) pertains to *a specific task* when evaluating LLM-generated codes. In our experiments, we might extend the use of dual@$k$ to gauge the efficiency of codes across *a dataset* by averaging the dual@$k$ values from all tasks within that dataset.

Next, we introduce how to construct $\mathcal{S}$ and $\mathcal{W}$, respectively.

### 4.1. Score matrix $\mathcal{S}$

We compute an independent pass@$k$ for each subtask $T_{ij}$ as its score $s_{ij}$. The pass@$k$ estimates the probability that at least one of $k$ sampled codes passes $T_{ij}$ by:

$$\text{pass}@k = 1 - \frac{\binom{n-c}{k}}{\binom{n}{k}}, \qquad (2)$$

where $n$ denotes the total number of codes generated by LLM, and $c$ is the number of codes passing the subtask. Note that our definition may differ slightly from Chen et al. (2021) in regards to the concept of 'pass': the original pass@$k$ does not account for the efficiency such that it suffices to compute correct solutions for all test cases; in our evaluation, a code is only considered to pass a subtask if it computes correct solutions *within both time and space limits* for all test cases.

### 4.2. Weight matrix $\mathcal{W}$

The construction of $\mathcal{W}$ is non-trivial because $\mathcal{W}$ is expected to represent how different ranks of subtasks contribute to dual@$k$. We write each component $\omega_{ij}$ of $\mathcal{W}$ as a *weight function* $\omega(i, j)$ of the indices $i$ and $j$. We impose the following considerations that induce constraints on $\omega(i, j)$.

**Monotonicity.** Subtasks $T_{ij}$ with greater $i$ or $j$ are guaranteed to be more challenging and therefore warrant higher weights. It gives the following monotonicity constraint for $\forall i \in [n]$ and $\forall j \in [m]$:

$$\omega(i+1, j) \geq \omega(i, j); \tag{3}$$
$$\text{and } \omega(i, j+1) \geq \omega(i, j). \tag{4}$$

**Convexity.** We allocate greater marginal rewards to codes that successfully navigate larger-scale subtasks, as they are more representative of the algorithm's complexity. Therefore, we ensure for $\forall i \in [n]$ and $\forall j \in [m]$:

$$\omega(i+1, j) - \omega(i, j) \geq \omega(i, j) - \omega(i-1, j); \tag{5}$$
$$\text{and } \omega(i, j+1) - \omega(i, j) \geq \omega(i, j) - \omega(i, j-1). \tag{6}$$

**Supermodularity.** We offer greater rewards to codes that excel in both time and space efficiency rather than those that only perform well in either of them. Formally, we ensure for $\forall i \in [n]$ and $\forall j \in [m]$:

$$\omega(i+1, j+1) + \omega(i, j) \geq \omega(i+1, j) + \omega(i, j+1). \tag{7}$$

The weight function $\omega(i, j)$ needs to satisfy Eqn. (3) to (7). Although there are numerous choices for $\omega$, we use a simple yet effective one by defining $\omega$ as:

$$\omega(i, j) = \tau^{i-1} \sigma^{j-1}, \tag{8}$$

where $\tau > 1, \sigma > 1$ are hyper-parameters weighing the importance of time and space, respectively. Theorem 4.1 indicates that Eqn. (8) satisfies all Eqn. (3) to (7), and its formal proof is deferred to Appendix E.

**Theorem 4.1.** *There exist $\tau > 1$ and $\sigma > 1$ such that the weight function defined by $\omega(i, j) = \tau^{i-1}\sigma^{j-1}$ satisfies Eqn. (3) to (7).*

**Remark on Eqn. (8).** We highlight that the weight function in Eqn. (8) makes our metric unified and adaptable for various application scenarios where time and space are valued differently. In situations where time is prioritized over space, we can establish $\tau > \sigma$. Conversely, if space is more critical, then $\tau < \sigma$. Additionally, our dual@$k$ can measure either time or space efficiency only by setting either $\sigma$ or $\tau$ to 0, respectively. Specifically, if we evaluate time efficiency only, it is sufficient to evaluate codes on subtasks $T_{i1}$ for $i \in [n]$ and set $\sigma = 0$. In this case, the weight function becomes $\omega(i, 1) = \tau^{i-1}$. So our work directly generalizes the previous ones that evaluate time efficiency only (Qiu et al., 2024; Du et al., 2024; Niu et al., 2024).

# 5. Evaluation

We comprehensively evaluate 50 popular LLMs (51 evaluated configurations, where DeepSeek-V3.2 is evaluated

with both standard and thinking modes), including 35 open-source models with a broad parameter spectrum from 6.7 to 685 billion and 15 closed-source models, on our BEST benchmark. Detailed model information is presented in Appendix F, Table 6. In our evaluation, we use C++ as the implementation language because its explicit memory management and lack of automatic runtime optimizations make differences in time and space efficiency more observable.

## 5.1. Experimental Setup

**Code generation by LLMs.** In our BEST, each task corresponds to multiple (typically 9) subtasks. For each subtask, we independently generate 20 codes from different LLMs. For open-source models, we use a temperature of 0.8 and a top_p value of 0.95 for sampling on a server equipped with 8 NVIDIA A100-80GB GPUs. For closed-source models, we access them via their official APIs and use the same sampling configuration.

**Code evaluation.** We run every LLM-generated code as well as our baselines on virtualized cloud servers hosted by Google Cloud (Ubuntu 22.04; Intel Xeon Platinum 8581C CPU@2.10GHz; G++ 11.4.0), under a standardized compiler setting, with no optimization flags enabled. We reiterate that dual@$k$ is device-independent after per-environment recalibration. Specifically, once the time and memory limits of subtasks are calibrated for a given execution environment using the corresponding baselines, the calculation of dual@$k$ does not directly use the measured runtime or memory usage of each generated code. This differs from existing metrics (Qiu et al., 2024; Huang et al., 2024; Du et al., 2024).

**Prompt design.** To reduce the influence of prompt engineering on efficiency improvement, we employ a uniform one-shot prompt for each task and baseline, as illustrated in Appendix F Figure 4.

**Hyper-parameters.** We consider dual@$k$ with $k = 1, 10$ in experiments. In the weight function $\omega(i, j) = \tau^{i-1}\sigma^{j-1}$, we consider different values (0 or 1.2) of $\tau$ and $\sigma$ to evaluate time efficiency, space efficiency, or both.

## 5.2. Experimental Results

**Experiment #1: time-space efficiency evaluation on LLMs.** Table 2 presents our evaluation on 50 LLMs using the dual@$k$ metric on our BEST. We compare to the correctness metric pass@$k$ (Chen et al., 2021; Liu et al., 2023a) as defined in Eqn. (2). Note that for pass@$k$, we measure how many of $k$ sampled codes can pass all test cases correctly, which is consistent with the definition in the literature. For time-only efficiency, we compare to the metric eff@$k$ (Qiu et al., 2024), whose details are presented in Appendix F. Our

*Table 2.* Code efficiency evaluation on 50 widely-studied LLMs. Evaluations are conducted on three different choices of $\tau$ and $\sigma$ for evaluation on either time or space, as well as their dual efficiency. The dual@$k$ metric is compared to the eff@$k$ (Qiu et al., 2024) and the pass@$k$ (Chen et al., 2021). The most efficient result for each metric is bolded and underlined.

| Model | Time Efficiency ($\tau = 1.2, \sigma = 0$) | | | Space Efficiency ($\tau = 0, \sigma = 1.2$) | | Time-Space Dual Efficiency ($\tau = 1.2, \sigma = 1.2$) | | | | | | | | pass@10 |
| | | | | | | Easy | | Medium | | Hard | | Overall | | |
| | dual@1 | dual@10 | eff@10 | dual@1 | dual@10 | dual@1 | dual@10 | dual@1 | dual@10 | dual@1 | dual@10 | dual@1 | dual@10 | |
|---|---|---|---|---|---|---|---|---|---|---|---|---|---|---|
| CodeLlama-34B-Hf | 0.030 | 0.108 | 0.166 | 0.015 | 0.063 | 0.042 | 0.158 | 0.007 | 0.033 | 0.001 | 0.004 | 0.011 | 0.046 | 0.155 |
| CodeLlama-34B-Instruct-Hf | 0.030 | 0.110 | 0.174 | 0.017 | 0.069 | 0.055 | 0.208 | 0.005 | 0.024 | 0.000 | 0.004 | 0.013 | 0.052 | 0.155 |
| CodeLlama-13B-Hf | 0.023 | 0.087 | 0.136 | 0.013 | 0.053 | 0.034 | 0.135 | 0.006 | 0.027 | 0.000 | 0.001 | 0.009 | 0.038 | 0.123 |
| CodeLlama-13B-Instruct-Hf | 0.030 | 0.097 | 0.137 | 0.016 | 0.057 | 0.048 | 0.155 | 0.006 | 0.026 | 0.001 | 0.004 | 0.012 | 0.043 | 0.131 |
| DeepSeek-V3.2-Thinking | 0.628 | **0.795** | **0.862** | 0.594 | **0.768** | 0.829 | 0.914 | 0.609 | **0.769** | 0.259 | **0.481** | 0.508 | **0.678** | **0.845** |
| DeepSeek-V3.2 | 0.613 | 0.736 | 0.783 | 0.588 | 0.718 | 0.834 | 0.878 | 0.640 | 0.747 | 0.233 | 0.404 | 0.510 | 0.630 | 0.784 |
| DeepSeek-R1 | **0.641** | 0.734 | 0.766 | **0.612** | 0.707 | **0.870** | **0.927** | 0.636 | 0.747 | **0.276** | 0.394 | **0.533** | 0.635 | 0.756 |
| DeepSeek-V3-0324 | 0.584 | 0.682 | 0.715 | 0.562 | 0.651 | 0.830 | 0.881 | **0.643** | 0.736 | 0.190 | 0.297 | 0.492 | 0.582 | 0.711 |
| DeepSeek-V3 | 0.527 | 0.650 | 0.708 | 0.528 | 0.629 | 0.816 | 0.895 | 0.595 | 0.714 | 0.119 | 0.218 | 0.441 | 0.544 | 0.707 |
| DeepSeek-Coder-V2-Lite-Instruct | 0.281 | 0.422 | 0.476 | 0.280 | 0.412 | 0.602 | 0.746 | 0.220 | 0.376 | 0.027 | 0.057 | 0.217 | 0.318 | 0.516 |
| DeepSeek-Coder-V2-Lite-Base | 0.284 | 0.443 | 0.537 | 0.267 | 0.419 | 0.606 | 0.807 | 0.203 | 0.384 | 0.015 | 0.041 | 0.206 | 0.327 | 0.547 |
| DeepSeek-Coder-33B-Instruct | 0.169 | 0.396 | 0.483 | 0.107 | 0.297 | 0.265 | 0.640 | 0.091 | 0.280 | 0.004 | 0.023 | 0.090 | 0.246 | 0.466 |
| DeepSeek-Coder-33B-Base | 0.156 | 0.363 | 0.467 | 0.095 | 0.259 | 0.241 | 0.585 | 0.081 | 0.237 | 0.004 | 0.018 | 0.081 | 0.216 | 0.430 |
| DeepSeek-Coder-6.7B-Instruct | 0.051 | 0.191 | 0.260 | 0.025 | 0.109 | 0.070 | 0.276 | 0.018 | 0.084 | 0.000 | 0.004 | 0.021 | 0.089 | 0.225 |
| DeepSeek-Coder-6.7B-Base | 0.046 | 0.162 | 0.233 | 0.020 | 0.075 | 0.058 | 0.205 | 0.014 | 0.059 | 0.000 | 0.003 | 0.018 | 0.065 | 0.189 |
| Qwen3-235B-A22B | 0.500 | 0.653 | 0.750 | 0.469 | 0.627 | 0.782 | 0.911 | 0.516 | 0.708 | 0.104 | 0.216 | 0.398 | 0.544 | 0.704 |
| Qwen3-Coder-30B-A3B-Instruct | 0.443 | 0.599 | 0.702 | 0.419 | 0.576 | 0.727 | 0.886 | 0.460 | 0.663 | 0.080 | 0.162 | 0.356 | 0.499 | 0.660 |
| QwQ-32B | 0.585 | 0.723 | 0.775 | 0.540 | 0.679 | 0.854 | 0.921 | 0.571 | 0.729 | 0.220 | 0.378 | 0.482 | 0.621 | 0.750 |
| Qwen2.5-Coder-32B | 0.402 | 0.583 | 0.669 | 0.355 | 0.534 | 0.685 | 0.886 | 0.392 | 0.636 | 0.030 | 0.089 | 0.300 | 0.459 | 0.644 |
| Qwen2.5-Coder-14B | 0.328 | 0.504 | 0.595 | 0.291 | 0.458 | 0.649 | 0.861 | 0.280 | 0.495 | 0.018 | 0.065 | 0.245 | 0.390 | 0.572 |
| Qwen2.5-Coder-7B | 0.260 | 0.420 | 0.511 | 0.215 | 0.371 | 0.547 | 0.770 | 0.184 | 0.377 | 0.007 | 0.025 | 0.184 | 0.310 | 0.488 |
| Yi-34B | 0.098 | 0.235 | 0.329 | 0.071 | 0.171 | 0.222 | 0.465 | 0.028 | 0.095 | 0.002 | 0.009 | 0.057 | 0.134 | 0.303 |
| Yi-9B | 0.079 | 0.227 | 0.307 | 0.045 | 0.140 | 0.127 | 0.363 | 0.027 | 0.097 | 0.001 | 0.003 | 0.036 | 0.112 | 0.278 |
| Llama-3.1-70B | 0.393 | 0.480 | 0.539 | 0.389 | 0.470 | 0.726 | 0.807 | 0.393 | 0.490 | 0.095 | 0.143 | 0.336 | 0.410 | 0.529 |
| Llama-3.1-8B | 0.169 | 0.244 | 0.306 | 0.170 | 0.248 | 0.398 | 0.524 | 0.134 | 0.214 | 0.006 | 0.015 | 0.134 | 0.194 | 0.305 |
| Mixtral-8x7B-V0.1 | 0.087 | 0.203 | 0.282 | 0.063 | 0.149 | 0.193 | 0.378 | 0.027 | 0.091 | 0.001 | 0.004 | 0.050 | 0.113 | 0.269 |
| Mixtral-8x7B-Instruct-V0.1 | 0.084 | 0.188 | 0.258 | 0.062 | 0.151 | 0.190 | 0.382 | 0.027 | 0.091 | 0.001 | 0.004 | 0.049 | 0.114 | 0.251 |
| StarCoder2-15B | 0.009 | 0.053 | 0.084 | 0.003 | 0.019 | 0.009 | 0.053 | 0.002 | 0.012 | 0.000 | 0.001 | 0.003 | 0.016 | 0.063 |
| StarCoder2-15B-Instruct-V0.1 | 0.058 | 0.195 | 0.269 | 0.031 | 0.115 | 0.084 | 0.292 | 0.015 | 0.064 | 0.001 | 0.007 | 0.023 | 0.087 | 0.257 |
| Mistral-7B-Instruct-V0.3 | 0.041 | 0.109 | 0.175 | 0.028 | 0.080 | 0.083 | 0.209 | 0.010 | 0.035 | 0.000 | 0.002 | 0.021 | 0.057 | 0.161 |
| CodeGemma-7B | 0.080 | 0.198 | 0.278 | 0.061 | 0.154 | 0.155 | 0.396 | 0.031 | 0.088 | 0.003 | 0.008 | 0.044 | 0.118 | 0.259 |
| WizardCoder-33B-V1.1 | 0.165 | 0.370 | 0.468 | 0.109 | 0.269 | 0.281 | 0.615 | 0.079 | 0.237 | 0.003 | 0.013 | 0.088 | 0.220 | 0.437 |
| Phind-CodeLlama-34B-V1 | 0.076 | 0.210 | 0.304 | 0.049 | 0.139 | 0.150 | 0.371 | 0.020 | 0.076 | 0.001 | 0.007 | 0.039 | 0.107 | 0.278 |
| Magicoder-S-CL-7B | 0.019 | 0.076 | 0.117 | 0.010 | 0.041 | 0.031 | 0.118 | 0.003 | 0.015 | 0.001 | 0.004 | 0.007 | 0.031 | 0.100 |
| StableLM-2-12B | 0.042 | 0.128 | 0.200 | 0.030 | 0.099 | 0.080 | 0.219 | 0.013 | 0.057 | 0.001 | 0.006 | 0.022 | 0.068 | 0.192 |
| XwinCoder-34B | 0.039 | 0.120 | 0.167 | 0.020 | 0.068 | 0.064 | 0.201 | 0.007 | 0.032 | 0.000 | 0.000 | 0.016 | 0.053 | 0.151 |
| **Closed-Source Models** | | | | | | | | | | | | | | |
| Gemini-3-Pro-Preview | **0.802** | **0.890** | **0.923** | 0.760 | **0.868** | 0.906 | **0.940** | 0.790 | **0.870** | **0.602** | **0.787** | **0.735** | **0.849** | **0.904** |
| Gemini-2.5-Pro | 0.685 | 0.825 | 0.863 | 0.619 | 0.796 | 0.748 | 0.904 | 0.690 | 0.867 | 0.405 | 0.597 | 0.586 | 0.764 | 0.853 |
| Gemini-2.5-Flash | 0.664 | 0.777 | 0.821 | 0.554 | 0.671 | 0.647 | 0.678 | 0.649 | 0.780 | 0.315 | 0.477 | 0.509 | 0.633 | 0.806 |
| GPT-5.2 | 0.786 | 0.855 | 0.898 | **0.766** | 0.841 | **0.907** | 0.926 | **0.798** | 0.853 | 0.570 | 0.717 | 0.725 | 0.811 | 0.879 |
| O4-Mini | 0.553 | 0.765 | 0.808 | 0.476 | 0.692 | 0.783 | 0.911 | 0.508 | 0.755 | 0.160 | 0.375 | 0.419 | 0.628 | 0.789 |
| GPT-4.1 | 0.532 | 0.650 | 0.690 | 0.517 | 0.627 | 0.808 | 0.888 | 0.544 | 0.670 | 0.175 | 0.275 | 0.443 | 0.549 | 0.688 |
| GPT-4o | 0.492 | 0.625 | 0.676 | 0.461 | 0.590 | 0.795 | 0.923 | 0.515 | 0.703 | 0.081 | 0.156 | 0.391 | 0.519 | 0.673 |
| GPT-4o-Mini | 0.370 | 0.496 | 0.562 | 0.359 | 0.476 | 0.724 | 0.858 | 0.372 | 0.525 | 0.022 | 0.058 | 0.297 | 0.398 | 0.568 |
| GPT-3.5-Turbo-1106 | 0.332 | 0.544 | 0.617 | 0.272 | 0.473 | 0.602 | 0.866 | 0.253 | 0.528 | 0.019 | 0.064 | 0.226 | 0.403 | 0.597 |
| Grok-3 | 0.482 | 0.614 | 0.669 | 0.499 | 0.615 | 0.783 | 0.890 | 0.516 | 0.655 | 0.125 | 0.208 | 0.407 | 0.516 | 0.695 |
| Grok-3-Mini | 0.466 | 0.584 | 0.620 | 0.426 | 0.534 | 0.816 | 0.907 | 0.482 | 0.634 | 0.084 | 0.154 | 0.384 | 0.489 | 0.608 |
| Claude-Sonnet-4-5-20250929 | 0.580 | 0.684 | 0.761 | 0.598 | 0.697 | 0.862 | 0.927 | 0.623 | 0.741 | 0.208 | 0.303 | 0.498 | 0.596 | 0.750 |
| Claude-3-7-Sonnet-20250219 | 0.465 | 0.638 | 0.700 | 0.450 | 0.645 | 0.683 | 0.882 | 0.451 | 0.669 | 0.121 | 0.230 | 0.360 | 0.529 | 0.720 |
| Claude-3-5-Sonnet-20241022 | 0.481 | 0.628 | 0.682 | 0.471 | 0.606 | 0.770 | 0.902 | 0.517 | 0.683 | 0.092 | 0.185 | 0.391 | 0.520 | 0.681 |
| Claude-3-Haiku-20240307 | 0.408 | 0.546 | 0.601 | 0.415 | 0.547 | 0.687 | 0.822 | 0.417 | 0.576 | 0.069 | 0.140 | 0.327 | 0.445 | 0.634 |

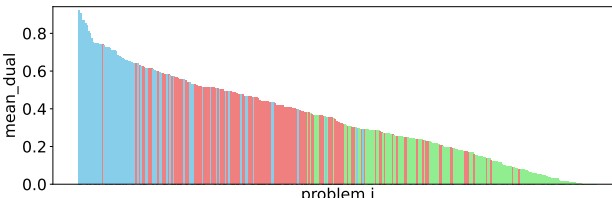

*Figure 2.* Distribution of task difficulties. Blue for 'easy', red for 'medium', and green for 'hard'. The tasks are placed in a decreasing order based on the average dual@10 among all LLMs.

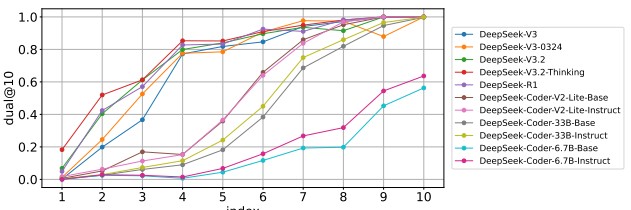

*Figure 3.* Efficiency (dual@10) on BEST of codes generated by DeepSeek family.

*Table 3.* The dual@10 ($\sigma = \tau = 1.2$) among 10 types of algorithms for three representative LLMs. The most efficient result under each difficulty label is bolded and underlined, while the least is highlighted in grey.

| Categorization | GPT-5.2 | | | | Gemini-3-Pro-Preview | | | | DeepSeek-V3.2-Thinking | | | |
|---|---|---|---|---|---|---|---|---|---|---|---|---|
| | Easy | Medium | Hard | Overall | Easy | Medium | Hard | Overall | Easy | Medium | Hard | Overall |
| String | 0.941 | 0.762 | 0.760 | 0.822 | **1.000** | 0.865 | 0.787 | 0.886 | 0.937 | 0.808 | 0.528 | 0.762 |
| Sort | **1.000** | 0.911 | 0.824 | 0.894 | **1.000** | 0.916 | 0.858 | 0.910 | 0.934 | 0.798 | 0.511 | 0.710 |
| Greedy | 0.940 | 0.844 | 0.867 | 0.872 | 0.941 | 0.880 | **0.944** | 0.922 | 0.931 | 0.776 | 0.623 | 0.727 |
| Graph | 0.994 | 0.930 | 0.614 | 0.763 | **1.000** | 0.938 | 0.730 | 0.830 | 0.982 | **0.879** | 0.468 | 0.664 |
| Data Structures | 0.839 | 0.837 | 0.712 | 0.768 | 0.818 | 0.853 | 0.760 | 0.797 | 0.788 | 0.645 | 0.377 | 0.514 |
| Two Pointers | 0.934 | **0.940** | **0.925** | **0.933** | 0.950 | **0.949** | 0.896 | **0.928** | 0.981 | 0.862 | **0.669** | **0.808** |
| Math | 0.869 | 0.891 | 0.742 | 0.819 | 0.868 | 0.901 | 0.784 | 0.841 | 0.883 | 0.785 | 0.497 | 0.680 |
| Search | **1.000** | 0.803 | 0.882 | 0.864 | **1.000** | 0.864 | 0.850 | 0.875 | **1.000** | 0.761 | 0.497 | 0.674 |
| DP | 0.952 | 0.868 | 0.678 | 0.758 | 0.924 | 0.907 | 0.757 | 0.816 | 0.885 | 0.824 | 0.468 | 0.610 |
| Other | **1.000** | 0.891 | 0.602 | 0.799 | **1.000** | 0.875 | 0.615 | 0.795 | 0.947 | 0.825 | 0.524 | 0.730 |
| Overall | 0.926 | 0.853 | 0.717 | 0.811 | 0.940 | 0.870 | 0.787 | 0.849 | 0.914 | 0.769 | 0.481 | 0.678 |

primary findings suggest that *although LLMs struggle with generating time-efficient code, they perform even poorer when it comes to generating space-efficient code*. Detailed findings include:

- For our time-space dual efficiency ($\tau = \sigma = 1.2$), we first observe that *dual@10 is consistently lower than pass@10 for all LLMs*, indicating the focus on code correctness possibly at the expense of code efficiency in existing LLM studies. Secondly, *for most LLMs, worse performances in correctness induce poorer efficiency, with some surprising exceptions.* For example, CodeGemma-7B has higher dual@10 but lower pass@10 than Phind-CodeLlama-34B-V1.

- For time-only efficiency, our dual@10 behaves roughly *consistent* with eff@10 in rankings of LLMs. However, in terms of time-only evaluation, our dual@$k$ may outperform eff@$k$ in both simplifying calculations and the flexibility of assigning different weights to subtasks.

- For space-only efficiency, we observe that *almost all LLMs are less effective at generating space-efficient compared to time-efficient algorithms*. One exception is Llama-3.1-8B, which has the time-only dual@10 of 0.244, slightly smaller than the space-only dual@10 of 0.248. It demonstrates that LLM developers may focus even less on space than on time of code efficiency. Additionally, *commercial models generally perform better than open-source models* regarding space efficiency, and the highest performance is achieved by Gemini-3-Pro-Preview. Finally and surprisingly, *LLMs that are better at generating time-efficient codes may behave worse in generating space-efficient codes*. An example is GPT-3.5-Turbo-1106 versus GPT-4o-Mini.

- For different levels of difficulty, the performance of the LLMs is *consistent* with the difficulty ratings of the tasks by the human experts, which demonstrates the

validity of our categorizations. Figure 2 presents the distribution of difficulty across all tasks in BEST. We plot the average dual@10's on all evaluated LLMs and rank them in decreasing order, which further validates such consistency.

- We are seeing *a narrowing of the performance gap between open-source and commercial models* in generating efficient code. For example, the highest dual@10 among open-source models is achieved by DeepSeek-V3.2-Thinking, which is higher than all models except Gemini-3-Pro-Preview, Gemini-2.5-Pro, and GPT-5.2.

- We observe that reasoning strategies, such as Chain-of-Thought (CoT) (Wei et al., 2022), help LLMs generate more efficient code. Large reasoning models, such as Gemini-3-Pro-Preview and DeepSeek-V3.2-Thinking, achieve significantly higher overall dual@10 values compared to other LLMs. This performance advantage may stem from their step-by-step reasoning, which helps them avoid intuitive yet inefficient implementations and produce solutions that better balance time and space efficiency.

- There is a clear diversity effect: pass@1 averages 0.351 compared to 0.495 for pass@10, and dual@1 averages 0.250 compared to 0.349 for dual@10, indicating that a temperature of 0.8 effectively explores the diversity of generated solutions (Chen et al., 2021). Interestingly, DeepSeek-R1 achieves a higher dual@1 than DeepSeek-V3.2-Thinking but is slightly worse on dual@10. This suggests that DeepSeek-R1 is more likely to produce an efficient solution on its first attempt, whereas DeepSeek-V3.2-Thinking benefits more from diverse sampling.

**Experiment #2: efficiency evaluation among different types of algorithms.** Table 3 presents our dual@10, with $\sigma = \tau = 1.2$, evaluated across 10 different types of algorithms. We evaluate three representative LLMs, including GPT-5.2 (closed-source), Gemini-3-Pro-Preview (top

performance), and DeepSeek-V3.2-Thinking (open-source). More results on other LLMs are deferred to Appendix G. We observe *significant differences* in the capability of different LLMs to generate efficient code towards different types of algorithms. For example, GPT-5.2 performs best in Two Pointers, while Gemini-3-Pro-Preview performs best in String. Gemini-3-Pro-Preview also performs well on Data Structures, whereas DeepSeek-V3.2-Thinking lags behind in this category. We occasionally observe slight reversals across difficulty tiers within a category, e.g., for GPT-5.2 on Search, the score on hard problems is slightly higher than on medium ones. This may be because solving flash-of-genius problems, such as Search and Greedy, sometimes relies solely on a single effective trick or insight: once the model discovers such an idea, even a harder problem can be solved efficiently, regardless of its nominal difficulty. Another finding suggests that *while LLMs excel at utilizing knowledge, they struggle with logical reasoning*. For example, all three LLMs excel in String, Sort, and Two Pointers, where the tasks are more structured and predefined. We believe that their solutions may have been included in the training data. For tricky categories such as Graph and DP, all three LLMs show relatively lower scores. The generated code, while efficient, may not fully capture the corner cases.

We believe that special attention to different types of algorithms is essential to evaluate the efficiency of LLM-generated codes. Including specific data on their underperforming algorithms in the training dataset or fine-tuning on some algorithm-specific tasks may enhance performance.

**Experiment #3: efficiency evaluation within a family of LLMs.** For a more fine-grained evaluation, Figure 3 presents the code efficiency generated by the DeepSeek family. We divide tasks into 10 groups, indexed from 1 to 10, based on the average dual@10 computed by all LLMs. Index 1 corresponds to the hardest tasks of dual@10 $\in [0, 0.1)$, index 2 corresponds to tasks of dual@10 $\in [0.1, 0.2)$, and so forth. We plot the average dual@10's among tasks in every single group. Our observation shows that the later released LLMs exhibit substantially improved capability in generating efficient code compared to earlier versions. Notably, DeepSeek-V3.2-Thinking achieves performance comparable to DeepSeek-R1 while incurring lower inference cost.

More experiments and findings, such as an evaluation on the optimization policy of LLMs, are deferred to Appendix G due to the page limitation.

## 6. Conclusion

This paper presents BEST, a thorough and high-quality benchmark for assessing the ability of LLMs to generate efficient code in terms of both time and space. BEST includes 440 expert-crafted coding tasks, each featuring subtasks of varying difficulty along with the Pareto optimal baselines. We have diligently sought suitable evaluation tasks from multiple online coding platforms, unlike prior works that either use unverified tasks from a single platform or select a limited subset from a known dataset. Additionally, we propose a subtask-based evaluation framework and present a novel and unified metric dual@$k$, which can be seamlessly adjusted for any scenario where time and space are valued differently. Our thorough evaluation indicates that while current LLMs struggle to produce time-efficient code, their understanding of space efficiency is even poorer. Interesting future directions include exploring ways to enhance LLM in efficient coding, either through prompt engineering or LLM fine-tuning based on established efficiency benchmarks.

## Acknowledgements

The work is supported in part by the Major Research Project of Hubei Province (2023BAA027), and in part by the National Natural Science Foundation of China (62572208). Qiankun Zhang is supported by the National Natural Science Foundation of China (Grant 62302183), Open Foundation of Key Laboratory of Cyberspace Security, Ministry of Education of China (Grant KLCS20240401), and CCF-DiDi GAIA Collaborative Research Funds (Grant CCF-DiDi GAIA 202522). Jing Wang is supported by the Nature Science Foundation of Hubei Province (Grant 2025DJA050, 2025AFD748). Bin Yuan is supported by the National Natural Science Foundation of China (Grant 62372191), the Open Topics from The Lion Rock Labs of Cyberspace Security (Grant LRL24013), and Songshan Laboratory (Grant 241110210200). Xianjun Deng is supported by the National Key R&D Program of China (Grant 2022YFE0138600), and the National Natural Science Foundation of China (Grant U24B20153).

## Impact Statement

The introduction of BEST as a rigorous benchmark for evaluating the capability of LLMs in generating efficient code has the potential to significantly influence the trajectory of machine learning research. By emphasizing efficiency in time and space, BEST provides a standardized framework that encourages researchers to develop models that go beyond traditional accuracy metrics, fostering innovation in optimization techniques and model architectures. This benchmark can serve as a catalyst for advancing the state of the art in resource-efficient AI, promoting research into trade-offs between computational efficiency and performance. Furthermore, BEST creates opportunities for cross-disciplinary collaboration between machine learning, software engineering, and systems research, uniting efforts to address challenges in scalability, sustainability, and real-

world applicability. By establishing a clear and measurable set of goals, BEST not only sets a new standard for evaluating LLMs but also inspires the community to consider the environmental and practical implications of their models, ultimately steering machine learning research toward a more efficient and responsible future.

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

## A. BEST Specification

Table 4 introduces each field included in our benchmark, BEST, along with a concise description. The details are given in Section 3. Table 5 shows the statistics under different labels.

*Table 4.* Task card in BEST.

| Field | Description |
|---|---|
| `task_statement` | A clear and concise statement of the coding problem, outlining the requirements and goals for the solution. |
| `interface_declaration` | The required input-output format, function signatures, and etc. |
| `example` | Example inputs and the corresponding outputs. |
| `subtasks` | Tasks specifying an input scale and time-space limits required to measure different levels of efficiency. |
| `test_case` | Examples of specific inputs and expected outputs that the generated code should handle correctly. |
| `baseline` | Expert-crafted efficient standard implementation (solution) for each subtask. |
| `categorization` | Types of algorithms that are used in baselines, e.g., dynamic programming, greedy and etc. |
| `difficulty` | "easy", "medium", or "hard". |
| `script` | Any additional scripts or code templates that are required for setting up the task, running test cases, or evaluating the performance of the generated code. |

*Table 5.* Statistics of BEST.

| Categorization | Easy | Medium | Hard | Total |
|---|---|---|---|---|
| String | 21 | 21 | 17 | 59 |
| Sort | 28 | 27 | 28 | 83 |
| Greedy | 23 | 36 | 49 | 108 |
| Graph | 6 | 23 | 22 | 51 |
| Data Structure | 9 | 20 | 36 | 65 |
| Two Pointers | 11 | 29 | 25 | 65 |
| Math | 40 | 44 | 45 | 129 |
| Search | 7 | 30 | 25 | 62 |
| DP | 7 | 26 | 54 | 87 |
| Other | 23 | 43 | 16 | 82 |
| Total | 111 | 176 | 153 | 440 |

## B. An Example: Range Sum Query

This section presents a sample task in our benchmark: range sum query, which is illustrated in Figure 1.

The task statement, interface declaration, and examples of the sample task are shown in Figure 1. In addition, it is categorized as a 'data structure' task with a difficulty level of 'medium'.

Details regarding baselines, subtasks, and test cases for this task are in the following subsections.

### B.1. Baselines

We provide three Pareto-optimal reference implementations, each with different time and space complexity trade-offs. The time complexities are $O(n^2)$, $O(n\sqrt{n})$, and $O(n \log n)$, while the corresponding space complexities are $O(1)$, $O(\sqrt{n})$, and $O(n)$. These examples demonstrate the common principle that reducing time complexity often necessitates increased auxiliary space for storing intermediate results, thereby accelerating query processing.

**Algorithm A: enumeration.**  For each query, directly enumerate and sum the elements in the range $[l, r]$. For an update, modify the element at the specified position. The time complexity is $O(n^2)$, and the extra space complexity is $O(1)$.

```cpp
vector<long long> solve(vector<int> &a, vector<array<int, 3>> &ops) {
    for (auto &[op, l, r] : ops) {
        if (op == 1) {
            a[l - 1] = r;
        } else {
            long long sum = 0;
            for (int i = l - 1; i < r; i++) {
                sum += a[i];
            }
            answer.emplace_back(sum);
        }
    }
    return answer;
}
```

**Algorithm B: sqrt decomposition.**  The array is divided into blocks of size approximately $\sqrt{n}$. For each block, a variable is used to maintain the sum of the integers within that block. For a range query, the sums of the blocks fully contained in the query range are added directly. If the range starts or ends in the middle of a block, the remaining elements are handled by summing them directly. It only needs to sum up at most $O(\sqrt{n})$ blocks and sum up at most $O(\sqrt{n})$ elements. Therefore, the time complexity is $O(n\sqrt{n})$, and the extra space complexity is $O(\sqrt{n})$.

```cpp
vector<long long> solve(vector<int> &a, vector<array<int, 3>> &ops) {
    int n = a.size();
    int blockSize = static_cast<int>(sqrt(n)) + 1;
    int blockCount = (n + blockSize - 1) / blockSize;
    vector<long long> blockSum(blockCount, 0);
    for (int i = 0; i < n; ++i) {
        blockSum[i / blockSize] += a[i];
    }
    for (auto &op : ops) {
        int type = op[0];
        if (type == 1) {
            int p = op[1] - 1;
            int b = op[2];
            int blockIndex = p / blockSize;
            blockSum[blockIndex] += b - a[p];
            a[p] = b;
        } else if (type == 2) {
            int l = op[1] - 1;
            int r = op[2] - 1;
            long long sum = 0;
            int startBlock = l / blockSize;
            int endBlock = r / blockSize;
            if (startBlock == endBlock) {
                for (int i = l; i <= r; ++i) {
                    sum += a[i];
                }
            } else {
                for (int i = l; i < (startBlock + 1) * blockSize; ++i) {
                    sum += a[i];
                }
                for (int i = startBlock + 1; i < endBlock; ++i) {
                    sum += blockSum[i];
                }
                for (int i = endBlock * blockSize; i <= r; ++i) {
                    sum += a[i];
                }
            }
            answer.emplace_back(sum);
        }
```

```
40          }
41      return answer;
42  }
```

**Algorithm C: binary indexed tree.** The array is maintained in a tree-like structure named *binary indexed tree* (Fenwick, 1994) that helps in efficiently computing prefix sums. Each update operation adjusts the structure to maintain correct sums, and range sum queries are answered by computing the difference between prefix sums. The tree allows both update and query operations to be performed in logarithmic time. The time complexity is $O(n \log n)$, and the extra space complexity is $O(n)$.

```cpp
1   vector<long long> solve(vector<int> &a, vector<array<int, 3>> &ops) {
2       int n = a.size();
3       vector<long long> b(n + 1);
4       auto add = [&](int i, int x) {
5           for (; i <= n; i += i & -i) {
6               b[i] += x;
7           }
8       };
9       auto ask = [&](int i) {
10          long long res = 0;
11          for (; i > 0; i -= i & -i) {
12              res += b[i];
13          }
14          return res;
15      };
16      for (int i = 0; i < n; i++) {
17          add(i + 1, a[i]);
18      }
19      for (auto &[op, l, r] : ops) {
20          if (op == 1) {
21              add(l, -a[l - 1]);
22              a[l - 1] = r;
23              add(l, r);
24          } else {
25              answer.emplace_back(ask(r) - ask(l - 1));
26          }
27      }
28      return answer;
29  }
```

### B.2. Subtasks and Test Cases

The range sum query problem has three different Pareto-optimal reference implementations. These implementations have varying time complexities – $O(n^2)$, $O(n\sqrt{n})$, and $O(n \log n)$ – and corresponding space complexities – $O(1)$, $O(\sqrt{n})$, and $O(n)$. Our goal is to evaluate the algorithms and strategies employed by LLM-generated code, such as whether in-place operations or auxiliary data structures are used.

To distinguish these implementations, we divide the problem into $n \times m$ subtasks, where $n = m = 3$. Each subtask is defined by varying input scales, time limits, and space limits, while keeping other aspects, such as task descriptions and implementation interfaces, identical. A code passes subtask $(i, j)$ if and only if its time and space complexities are better than the subtask's baseline algorithm.

Intuitively, the first dimension $i \in [n]$ is designed to differentiate algorithms (or codes) based on their time complexities. We set the input scales and time limits as follows:

- $n = 10^4$, time limit = 500 ms,

- $n = 5 \cdot 10^4$, time limit = 1500 ms,

- $n = 10^5$, time limit = 2000 ms.

This is because a computer can perform around $10^8 \sim 10^9$ addition operations per second; therefore, we select the largest input scale that can be passed for each level of complexity. Besides, these subtasks are configured with rigorous test cases to ensure that they can distinguish between the $O(n^2)$, $O(n\sqrt{n})$, and $O(n \log n)$ implementations.

Similarly, the second dimension $j \in [m]$ is designed to differentiate codes with different space complexities. We set the memory limits as follows:

- memory limit = 6400 KB,

- memory limit = 640 KB,

- memory limit = 40 KB.

Each subtask also has a set of rigorous test cases to filter inefficient solutions. These time and memory limits are intentionally set as generous upper bounds (e.g., 10x slower than baselines) to avoid conflating algorithmic efficiency with hardware differences. The scale of the test data is large enough that an inefficient code would hardly meet the time limit, regardless of the hardware used. In addition, to ensure fairness, the evaluations are conducted with a fixed hardware configuration.

## C. Baseline Construction and Test Case Generation

**Baseline Construction.** To ensure the rigor of the standard reference implementations (baselines), we follow a structured four-stage process:

1. **Independent implementation**. Each author was randomly assigned a set of tasks. Tasks were often allocated to experts familiar with the specific problem category.

2. **Adversarial testing**. The authors created adversarial data generators or hand-crafted corner cases to validate the correctness and robustness of the implementations.

3. **Cross-review**. Each code implementation was reviewed by at least two non-author experts to ensure correctness and further checked whether it achieved Pareto optimality.

4. **Performance benchmarking**. For tasks sourced from online judge platforms like LeetCode and Codeforces, we verified our implementations against solutions in the leaderboard (with the least execution time or memory usage). This process also ensured that our solutions are (Pareto) optimal.

This multistage process ensures that our baselines are robust and reliable for evaluating LLM-generated code.

**Test Case Generation.** Our test cases were generated by experienced algorithm/programming contest participants and judges, following a rigorous process to ensure correctness and completeness.

For correctness:

- For problems sourced from online platforms, our implementations are verified via online judges to ensure correctness before generating test cases.

- For original problems, multiple experts write solutions independently. The solutions are validated via adversarial test cases generated by scripts or hand-crafted corner cases.

For completeness:

- If possible, we use all test cases in the input space.

- For other tasks, we hand-craft small and challenging test cases and generate large enough test cases for testing time and space efficiency.

- In addition, some test cases are derived from online judge platforms.

- We also craft targeted "hack" test cases to address common errors in invalid solutions.

For reproducibility, complete baselines and test cases have been made publicly available.

## D. Other Related Works

**Code generation via LLMs.** LLMs have made significant strides in code generation (Simon, 1963), driven by advancements in natural language processing (NLP), transformer models (Vaswani et al., 2017), and their application to code generation (Mastropaolo et al., 2021). These models have been trained on massive datasets that include both natural language and code, allowing them to generate code from textual descriptions. Several specialized code generation models have been developed to improve performance across various coding tasks. Notable models include Codex (Chen et al., 2021), Code Llama (Rozière et al., 2023), AlphaCode (Li et al., 2022), CodeGen (Nijkamp et al., 2022), Copilot (Microsoft, 2024), CodeT5+ (Wang et al., 2023b), PaLM-Coder (Chowdhery et al., 2022), InCoder (Fried et al., 2023), StarCoder (Lozhkov et al., 2024), SantaCoder (Allal et al., 2023), Qwen (Bai et al., 2023), and DeepSeek Coder (Guo et al., 2024; DeepSeek-AI et al., 2024). Contrastingly, models such as Codex (Chen et al., 2021) and Code Llama (Rozière et al., 2023) are fine-tuned from pre-existing foundation models like GPT3 (Brown et al., 2020) and Llama (Touvron et al., 2023b) on large-scale data, including open-source code repositories. In addition, general LLMs, such as GPT (OpenAI, 2023b;a), Llama (Touvron et al., 2023b;a; Meta, 2024), Claude (Anthropic, 2024a;b), Gemini (Google, 2023), and Mixtral (Jiang et al., 2024), also exhibit a range of abilities, from basic code generation to more complex tasks such as debugging (Prenner et al., 2022; Sobania et al., 2023).

**Correctness Evaluation on LLM-generated code.** Existing code generation benchmarks, such as HumanEval (Chen et al., 2021) and MBPP (Austin et al., 2021), Spider (Yu et al., 2018), and APPS (Hendrycks et al., 2021) have traditionally focused on evaluating functional correctness, among which, HumanEval (Chen et al., 2021) and MBPP (Austin et al., 2021) are the most prominent benchmarks, both centered on Python programming tasks. Recent works have aimed to enhance these benchmarks by addressing limitations in the testing mechanisms. For instance, HumanEval+ (Liu et al., 2023a) improves HumanEval by refining the test cases, thereby mitigating the problem of mistakenly accepting faulty solutions. ReCode (Wang et al., 2023a) approaches this issue by altering function names and docstrings to make the benchmark more adaptable and resistant to biases in generated solutions. In addition, ICPC-Eval (Xu et al., 2025) and LiveCodeBench Pro (Zheng et al., 2025) evaluate models' reasoning and problem-solving abilities on challenging competitive programming tasks, leveraging curated contest problems and primarily assessing solution correctness rather than efficiency. Moreover, HumanEval-X (Zheng et al., 2023), MultiPL-e (Cassano et al., 2023), MBXP (Athiwaratkun et al., 2023), and ODEX (Wang et al., 2022) extend HumanEval and MBPP beyond Python, introducing new problems across different programming languages or natural languages, further promoting the versatility of code generation evaluation. Other code evaluation benchmarks are dedicated to software engineering (Patil et al., 2023; Nijkamp et al., 2022; Liu et al., 2023b; Jimenez et al., 2024; Shrivastava et al., 2023; Zhang et al., 2023; Ding et al., 2022), data science (Lai et al., 2022; Yin et al., 2023; Zan et al., 2022; Jain et al., 2022), security (Siddiq & Santos, 2022), and so forth. Finally, there are some other works, including DeepPERF (Garg et al., 2022), PIE (Madaan et al., 2023) and SUPERSONIC (Chen et al., 2023) aiming at optimizing code performance in efficiency.

## E. Omitted Proofs

### E.1. Proof of Theorem 4.1

Let $\omega(i,j) = \tau^{i-1}\sigma^{j-1}$, and $\tau, \sigma > 1$.

**Proof of monotonicity.** Eqn. (3) holds because

$$\omega(i,j) = \tau^{i-1}\sigma^{j-1} \tag{9}$$

$$\leq \tau \cdot \tau^{i-1}\sigma^{j-1} \qquad (\tau > 1) \tag{10}$$

$$\leq \tau^{i}\sigma^{j-1} = \omega(i+1,j). \qquad (\text{definition of } \omega(i,j)) \tag{11}$$

Due to the symmetry of $\omega$, Eqn. (4) holds by interchanging $\tau$ and $\sigma$ in the proof of Eqn. (3).

**Proof of convexity.** Eqn. (5) holds because

$$\omega(i,j) - \omega(i-1,j) = \tau^{i-1}\sigma^{j-1} - \tau^{i-2}\sigma^{j-1} \tag{12}$$

$$\leq \tau \cdot \left(\tau^{i-1}\sigma^{j-1} - \tau^{i-2}\sigma^{j-1}\right) \qquad (\tau > 1) \tag{13}$$

$$\leq \tau^{i}\sigma^{j-1} - \tau^{i-1}\sigma^{j-1} = \omega(i+1,j) - \omega(i,j). \qquad (\text{definition of } \omega(i,j)) \tag{14}$$

Due to the symmetry of $\omega$, Eqn. (6) holds by interchanging $\tau$ and $\sigma$ in the proof of Eqn. (5).

**Proof of supermodularity.** To prove Eqn. (7), it suffices to prove $\omega(i+1, j+1) - \omega(i+1, j) \geq \omega(i, j+1) - \omega(i, j)$. Because

$$\omega(i, j+1) - \omega(i, j) = \tau^{i-1}\sigma^j - \tau^{i-1}\sigma^{j-1} \tag{15}$$

$$\leq \tau \cdot \left(\tau^{i-1}\sigma^j - \tau^{i-1}\sigma^{j-1}\right) \qquad (\tau > 1) \tag{16}$$

$$\leq \tau^i\sigma^j - \tau^i\sigma^{j-1} = \omega(i+1, j+1) - \omega(i+1, j), \qquad (\text{definition of } \omega(i, j)) \tag{17}$$

Eqn. (7) holds.

# F. More Experimental Details

## F.1. Model Details

We evaluate 35 open-source models and 15 closed-source models in code generation on our BEST. Please refer to Table 6 for details. We select widely used representative models such as GPT-4o (OpenAI, 2023a), Llama-3.1 (Meta, 2024), Claude (Anthropic, 2024b; 2025), DeepSeek-V3 and DeepSeek-V3.2 (Liu et al., 2024a; 2025), Qwen (Bai et al., 2023; Yang et al., 2024; 2025), and Yi (Young et al., 2024). We further include code-specific models tailored for coding expertise, such as CodeLlama (Roziere et al., 2023), DeepSeek-Coder (DeepSeek-AI et al., 2024), Qwen-Coder (Hui et al., 2024), StarCoder (Lozhkov et al., 2024). With the recent developments in reasoning models, we also include QwQ (Team, 2025), DeepSeek-R1 (Guo et al., 2025), GPT-5.2 (OpenAI, 2025), Gemini-2.5 (Comanici et al., 2025), etc., for comparison. We have reviewed the licenses of all involved models and ensured that they are available for non-commercial research purposes.

## F.2. Prompts for Code Generation

To ensure a standardized and informative presentation, we showcase a dedicated prompt for code generation using the LLM. As shown in Figure 4, the prompt consists of two main parts: the problem description and the code prompt. The problem description encompasses a detailed introduction to the problem, input-output examples, and time and space constraints. The code prompt specifies the exact format of the code.

## F.3. Calculation on dual@$k$

The value of dual@$k$ is defined by Eqn. (1). Algorithm 1 gives an implementation to calculate the value in practice.

---

**Algorithm 1** Calculate dual@$k$

---

**Require:** task $task$, evaluated codes $eval\_codes$, expert-crafted baselines $std\_codes$, parameter $k$, weights $\tau$ and $\sigma$, dimensions $n$ and $m$
**Ensure:** dual@k $dual\_at\_k$
 1: Initialize:
 2:     $W[i][j] \leftarrow \tau^{i-1} \cdot \sigma^{j-1}$ for all $i \in [1, n], j \in [1, m]$
 3:     $S_{eval}[n][m], S_{std}[n][m] \leftarrow 0$
 4: **for** $i \leftarrow 1$ to $n$ **do**
 5:     **for** $j \leftarrow 1$ to $m$ **do**
 6:         // compute pass@k for evaluated codes
 7:         $S_{eval}[i][j] \leftarrow \text{pass\_at\_k}\big(\text{count\_pass\_subtask}(eval\_codes, task[i][j])\big)$
 8:         //check if expert-crafted baselines pass the subtask
 9:         $S_{std}[i][j] \leftarrow \begin{cases} 1 & \text{if any\_pass\_subtask}(std\_codes, task[i][j]) \\ 0 & \text{otherwise} \end{cases}$
10:     **end for**
11: **end for**
12: // compute the ratio of the overall scores
13: $dual\_at\_k \leftarrow \dfrac{\langle S_{eval}, W \rangle}{\langle S_{std}, W \rangle}$

---

*Table 6.* Model Information.

| Model & Version | Scale | License / Copyright / Terms | Hyperlink |
|---|---|---|---|
| CodeLlama-34B-Hf | 34B | Llama 2 Community License Agreement | https://huggingface.co/codellama/CodeLlama-34b-hf |
| CodeLlama-34B-Instruct-Hf | 34B | Llama 2 Community License Agreement | https://huggingface.co/codellama/CodeLlama-34b-Instruct-hf |
| CodeLlama-13B-Hf | 13B | Llama 2 Community License Agreement | https://huggingface.co/codellama/CodeLlama-13b-hf |
| CodeLlama-13B-Instruct-Hf | 13B | Llama 2 Community License Agreement | https://huggingface.co/codellama/CodeLlama-13b-Instruct-hf |
| DeepSeek-V3.2 | 685B | MIT License | https://huggingface.co/deepseek-ai/DeepSeek-V3.2 |
| DeepSeek-R1 | 685B | MIT License | https://huggingface.co/deepseek-ai/DeepSeek-R1 |
| DeepSeek-V3-0324 | 685B | MIT License | https://huggingface.co/deepseek-ai/DeepSeek-V3-0324 |
| DeepSeek-V3 | 685B | DeepSeek License Agreement | https://huggingface.co/deepseek-ai/DeepSeek-V3 |
| DeepSeek-Coder-V2-Lite-Instruct | 16B | DeepSeek License Agreement | https://huggingface.co/deepseek-ai/DeepSeek-Coder-V2-Lite-Instruct |
| DeepSeek-Coder-V2-Lite-Base | 16B | DeepSeek License Agreement | https://huggingface.co/deepseek-ai/DeepSeek-Coder-V2-Lite-Base |
| DeepSeek-Coder-33B-Instruct | 33B | DeepSeek License Agreement | https://huggingface.co/deepseek-ai/deepseek-coder-33b-instruct |
| DeepSeek-Coder-33B-Base | 33B | DeepSeek License Agreement | https://huggingface.co/deepseek-ai/deepseek-coder-33b-base |
| DeepSeek-Coder-6.7B-Instruct | 6.7B | DeepSeek License Agreement | https://huggingface.co/deepseek-ai/deepseek-coder-6.7b-instruct |
| DeepSeek-Coder-6.7B-Base | 6.7B | DeepSeek License Agreement | https://huggingface.co/deepseek-ai/deepseek-coder-6.7b-base |
| Qwen3-235B-A22B | 235B | Apache-2.0 License | https://huggingface.co/Qwen/Qwen3-235B-A22B |
| Qwen3-Coder-30B-A3B-Instruct | 30B | Apache-2.0 License | https://huggingface.co/Qwen/Qwen3-Coder-30B-A3B-Instruct |
| QwQ-32B | 32B | Apache-2.0 License | https://huggingface.co/Qwen/QwQ-32B |
| Qwen2.5-Coder-32B | 32B | Apache-2.0 License | https://huggingface.co/Qwen/Qwen2.5-Coder-32B |
| Qwen2.5-Coder-14B | 14B | Apache-2.0 License | https://huggingface.co/Qwen/Qwen2.5-Coder-14B |
| Qwen2.5-Coder-7B | 7B | Apache-2.0 License | https://huggingface.co/Qwen/Qwen2.5-Coder-7B |
| Yi-34B | 34B | Apache-2.0 License | https://huggingface.co/01-ai/Yi-34B |
| Yi-9B | 9B | Apache-2.0 License | https://huggingface.co/01-ai/Yi-9B |
| Llama-3.1-70B | 70B | Llama 3.1 Community License Agreement | https://huggingface.co/meta-llama/Llama-3.1-70B |
| Llama-3.1-8B | 8B | Llama 3.1 Community License Agreement | https://huggingface.co/meta-llama/Llama-3.1-8B |
| Mixtral-8x7B-V0.1 | 8x7B | Apache-2.0 License | https://huggingface.co/mistralai/Mixtral-8x7B-v0.1 |
| Mixtral-8x7B-Instruct-V0.1 | 8x7B | Apache-2.0 License | https://huggingface.co/mistralai/Mixtral-8x7B-Instruct-v0.1 |
| StarCoder2-15B | 15B | BigCode OpenRAIL-M License | https://huggingface.co/bigcode/starcoder2-15b |
| StarCoder2-15B-Instruct-V0.1 | 15B | BigCode OpenRAIL-M License | https://huggingface.co/bigcode/starcoder2-15b-instruct-v0.1 |
| Mistral-7B-Instruct-V0.3 | 7B | Apache-2.0 License | https://huggingface.co/mistralai/Mistral-7B-Instruct-v0.3 |
| CodeGemma-7B | 7B | Gemma Terms of Use | https://huggingface.co/google/codegemma-7b |
| WizardCoder-33B-V1.1 | 33B | Microsoft Research License Terms | https://huggingface.co/WizardLMTeam/WizardCoder-33B-V1.1 |
| Phind-CodeLlama-34B-V1 | 34B | Llama 2 Community License Agreement | https://huggingface.co/Phind/Phind-CodeLlama-34B-v1 |
| Magicoder-S-CL-7B | 7B | Llama 2 Community License Agreement | https://huggingface.co/ise-uiuc/Magicoder-S-CL-7B |
| StableLM-2-12B | 12B | Stability AI Community License | https://huggingface.co/stabilityai/stablelm-2-12b |
| XwinCoder-34B | 34B | Llama 2 Community License Agreement | https://huggingface.co/Xwin-LM/XwinCoder-34B |
| Gemini-3-Pro-Preview | - | Google APIs Terms of Service | https://gemini.google.com |
| Gemini-2.5-Pro | - | Google APIs Terms of Service | https://gemini.google.com |
| Gemini-2.5-Flash | - | Google APIs Terms of Service | https://gemini.google.com |
| GPT-5.2 | - | OpenAI Service Terms | https://chatgpt.com |
| O4-Mini | - | OpenAI Service Terms | https://chatgpt.com |
| GPT-4.1 | - | OpenAI Service Terms | https://chatgpt.com |
| GPT-4o | - | OpenAI Service Terms | https://chatgpt.com |
| GPT-4o-Mini | - | OpenAI Service Terms | https://chatgpt.com |
| GPT-3.5-Turbo-1106 | - | OpenAI Service Terms | https://chatgpt.com |
| Grok-3 | - | xAI Terms of Service | https://grok.com |
| Grok-3-Mini | - | xAI Terms of Service | https://grok.com |
| Claude-Sonnet-4-5-20250929 | - | Anthropic Terms of Service | https://claude.ai |
| Claude-3-7-Sonnet-20250219 | - | Anthropic Terms of Service | https://claude.ai |
| Claude-3-5-Sonnet-20241022 | - | Anthropic Terms of Service | https://claude.ai |
| Claude-3-Haiku-20240307 | - | Anthropic Terms of Service | https://claude.ai |

## F.4. Calculation on eff@$k$

The metric, eff@$k$, is proposed to evaluate the time efficiency of code generated by LLMs, which is first introduced by Qiu et al. (2024). In this paper, to evaluate time efficiency only, we adapt eff@$k$ to our evaluation framework. Specifically, for each task, the value of eff@$k$ is calculated on subtasks $T_{i1}$ for $i \in [n]$. Let $L_i$ denote the time limit for $T_{i1}$, $t_{ir}^*$ denote the runtime on the $r$-th test case in $T_{i1}$, and $t_{irc}$ denote the runtime of code $c$ generated by LLM on the $r$-th test case in $T_{i1}$. If $c$ can not pass $T_{i1}$, set $t_{cr} = L_i$. Then, when sample $k$ codes from LLM, eff@$k$ for this task is defined as

$$\text{eff@}k = \frac{1}{\sum_{i=1}^{n} h_i} \mathbb{E} \max_c \left[ \sum_{i=1}^{n} h_i \cdot \frac{L_i - \max_r\{t_{irc}\}}{L_i - \max_r\{t_{ir}^*\}} \right], \tag{18}$$

where $h_i$ is hyperparameters represent the hardness of $T_{i1}$, which is intrinsic to the benchmark. In this paper, we set $h_i = \omega(i, 1)$.

Intuitively, Eqn. (18) uses a weighted sum method to sum the scores of all subtasks, where the score is the maximum runtime among all test cases within the subtask, and uses the runtime of baseline for normalization. We also abuse the notation of eff@$k$ to evaluate the performance of LLMs, by averaging the values of eff@$k$ among all tasks.

```
1    ### Question
2    You are a C++ expert.
3    Your task is to generate efficient code in both time and space.
4    You are given an array with n  integers (a₁,a₂,...,aₙ) and m operations to
5    perform on the array.
6    Each operation is one of following:
7        1. p,b: update the integer at position p to b, i.e., set aₚ = b.
8        2. l,r: query the sum of integers in the range [l,r], i.e.,∑ᵢ₌ₗʳ aᵢ.
9
10   Example:
11       Input: a = [1, 2, 3, 4, 5]
12             ops = [[2, 1, 3], [1, 2, 6], [2, 1, 3]]
13       Output: [6, 10]
14
15   Constraints:
16   1 ≤ n, m ≤ @data_scale
17   Time limit: @time_limit ms
18   Memory limit: @memory_limit KB
19
20   ### Code Prompt
21   class Solution {
22   public:
23       vector<long long> solve(vector<int> &a, vector<array<int, 3>> &ops) {
24           // write your code here
25       }
26   };
```

*Figure 4.* Prompts designed for code generation. Line 1-18 show the problem description, including detailed description, input-output examples, and time and space constraints. Line 20-26 show the exact format of the code.

## G. More Experiments

### G.1. Evaluation of Optimization Policy

We aim to conduct experiments on the ENAMEL (licensed under the Apache-2.0 License, and available for non-commercial research purposes) and BEST datasets to enhance the performance of code generated by LLMs through Supervised Fine-Tuning (SFT) and Direct Preference Optimization (DPO). The primary objective of the experiment is to examine the impact of fine-tuning on the time and space efficiency of code generated by LLMs, utilizing datasets from two different benchmarks.

**Experiment setup.**    A difficulty-balanced subset of the 440 problems is randomly selected to form the BEST-train datasets, and the remaining part of the problem is formed as the BEST-eval datasets. Similarly, we obtained the ENAMEL-train dataset. For each LLM, We evaluate the models' performance on BEST-eval datasets in terms of time efficiency and space efficiency and compare the effects of SFT and DPO by using ENAMEL-train and BEST-train datasets. The time and space efficiency is measured using the dual@10 metric, calculated for cases where $\tau = 1.2, \sigma = 0$ and $\tau = 0, \sigma = 1.2$. SFT fine-tunes the model on optimal solutions to improve its ability to generate functionally correct and efficient code by learning from task descriptions paired with the best-performing solutions. For SFT, we select the optimal solutions for each problem according to different benchmarks, with the training data formatted as

`<problem_description, prompts, optimal_solution>`. DPO optimizes the model to prefer efficient solutions by training on pairs of solutions with significant differences, directly aligning the model's preferences with efficiency goals. For DPO, we select pairs of solutions with the most significant differences according to different benchmarks, with the training data formatted as `<problem_description, prompts, better_solution, worse_solution>`. We apply LoRA in both SFT and DPO experiments. The AdamW optimizer is used with learning rates of 5e-5 for SFT and 3e-5 for DPO. SFT models are trained for 300 steps, while DPO models run for 600 steps.

**Experimental result.** As shown in Table 7, we investigate the impact of using different datasets and optimization methods on the time and space efficiency of the generated code. From the perspective of different optimization policies, DPO generally provides stable improvements in the time or space efficiency of the generated code, whereas SFT shows unstable effects. Additionally, when using DPO, we observe that compared to different datasets, the ENAMEL-train dataset can achieve a notable improvement in time efficiency, but has almost no impact on space efficiency. In contrast, the BEST-train dataset improves both time and space efficiency simultaneously. This observation aligns with the main focus of our benchmark: optimizing both the time and space efficiency of the code, whereas ENAMEL only considers time efficiency.

*Table 7.* The time and space efficiency obtained by applying SFT and DPO with ENAMEL-train and BEST-train datasets. The most efficient result of each model for each metric is bolded and underlined.

| Model | ENAMEL-train | | BEST-train | |
|---|---|---|---|---|
| | Time Efficiency | Space Efficiency | Time Efficiency | Space Efficiency |
| **CodeLlama-13B-Instruct-Hf** | - | - | - | - |
| +SFT | +0.012 | +0.003 | +0.016 | +0.032 |
| +DPO | **+0.022** | +0.001 | +0.012 | **+0.041** |
| **DeepSeek-Coder-33B-Base** | - | - | - | - |
| +SFT | -0.010 | -0.007 | -0.003 | -0.001 |
| +DPO | **+0.031** | -0.001 | +0.029 | **+0.028** |
| **Qwen2.5-Coder-14B** | - | - | - | - |
| +SFT | +0.002 | -0.004 | +0.008 | +0.007 |
| +DPO | +0.011 | +0.001 | **+0.014** | **+0.017** |
| **Yi-34B** | - | - | - | - |
| +SFT | -0.004 | -0.007 | 0.000 | -0.002 |
| +DPO | +0.018 | -0.006 | **+0.024** | **+0.033** |
| **Llama-3.1-70B** | - | - | - | - |
| +SFT | +0.011 | -0.009 | +0.016 | +0.009 |
| +DPO | **+0.028** | +0.001 | +0.025 | **+0.019** |
| **Mixtral-8x7B-V0.1** | - | - | - | - |
| +SFT | +0.021 | -0.004 | +0.022 | +0.001 |
| +DPO | **+0.042** | +0.002 | +0.038 | **+0.022** |
| **StarCoder2-15B-Instruct-V0.1** | - | - | - | - |
| +SFT | +0.002 | +0.001 | -0.003 | +0.007 |
| +DPO | **+0.028** | -0.002 | +0.026 | **+0.021** |

## G.2. Prompt Strategies

We conduct experiments with a few representative models, testing them with the original prompt as well as prompts explicitly emphasizing the optimization of time, space, and both time and space. The results are presented in Table 8, and we observe slight performance improvements across these prompts. These minor improvements could be attributed to the stronger emphasis on time and space constraints in the modified prompts.

## G.3. Cross-Language Consistency

To assess the robustness of BEST across programming languages and address potential language bias, we develop a Python instantiation of BEST and recalibrate time/memory limits based on Python baselines. For the cross-language analysis, we focus on a randomly sampled subset of 100 tasks. We then evaluate three representative models, Gemini-2.5-Pro, GPT-5.2, and DeepSeek-V3.2-Thinking, on this subset in both C++ and Python.

*Table 8.* Code efficiency evaluation on three representative LLMs. Evaluations are conducted on three choices of $\tau$ and $\sigma$. The codes are generated with optimized prompts that explicitly emphasize the optimization of time, space, and both time and space, compared against the results with the original prompt (see Figure 4).

| Model | Time Efficiency ($\tau = 1.2, \sigma = 0$, dual@10) | | Space Efficiency ($\tau = 0, \sigma = 1.2$, dual@10) | | Time-Space Dual Efficiency ($\tau = 1.2, \sigma = 1.2$, dual@10) | |
|---|---|---|---|---|---|---|
| | Original | Optimized | Original | Optimized | Original | Optimized |
| DeepSeek-V3 | 0.650 | 0.675 | 0.629 | 0.666 | 0.544 | 0.581 |
| Llama-3.1-70B | 0.480 | 0.487 | 0.470 | 0.479 | 0.410 | 0.416 |
| Qwen2.5-Coder-32B | 0.583 | 0.602 | 0.534 | 0.547 | 0.459 | 0.472 |

As shown in Table 9, although there are variations in absolute scores, the relative rankings of the models remain highly consistent. For instance, the Python Time Efficiency scores are slightly higher, possibly because interpreter overhead partially masks fine-grained algorithmic gaps relative to the baseline. In general, these results suggest that BEST and dual@$k$ can be instantiated across languages without substantially changing the conclusions. They provide initial evidence that our C++-based findings reflect general trends in efficient code generation, although extending BEST to additional languages remains an important direction for future work.

*Table 9.* Comparison of dual@10 scores for C++ and Python instantiations on a subset of BEST. The relative rankings of the models remain identical across languages.

| Model | Time Efficiency | | Space Efficiency | | Time-Space Dual Efficiency | |
|---|---|---|---|---|---|---|
| | C++ | Python | C++ | Python | C++ | Python |
| GPT-5.2 | 0.867 | 0.891 | 0.853 | 0.809 | 0.823 | 0.796 |
| Gemini-2.5-Pro | 0.841 | 0.858 | 0.811 | 0.774 | 0.776 | 0.751 |
| DeepSeek-V3.2-Thinking | 0.806 | 0.834 | 0.779 | 0.737 | 0.691 | 0.696 |

### G.4. Choice of $\tau$ and $\sigma$: Ablation

The entries of the weight matrix are controlled by the hyperparameters $\tau$ and $\sigma$. When evaluating time-space dual efficiency, we do not impose a preference between time and space, and therefore set $\tau = \sigma$. This leaves a single hyperparameter to choose. Intuitively, $\tau$ determines how quickly the weight $\omega(i, j)$ increases with the subtask rank, and thus controls how strongly the metric rewards solutions that pass harder subtasks. Our default choice is $\tau = 1.2$, mainly to make the numerical scale of dual@$k$ comparable to the prior time-efficiency metric eff@$k$ proposed by Qiu et al. (2024). Specifically, we vary $\tau$ from 1 to 1.5 and compare the resulting weights along the time-efficiency axis, i.e., $\omega(i, 1)$, with the normalized hardness hyperparameters $h_i$ used in Qiu et al. (2024). As shown in Figure 5, $\tau = 1.2$ gives a close match and is therefore used as the default setting.

We further evaluate the sensitivity of dual@$k$ to different choices of $\tau$ and $\sigma$. As reported in Table 10, changing these hyperparameters affects the absolute metric values, but the relative rankings of models remain stable. This indicates that our main empirical findings and model-level comparisons are not driven by a particular hyperparameter choice.

### G.5. Cross-Environment and Compiler-Flag Stability

To further examine whether the conclusions of BEST are sensitive to execution environments and compiler settings, we conduct an additional cross-environment study on the full benchmark. Specifically, we evaluate the top-5 models under four settings: Intel Xeon without compiler optimization flags, Intel Xeon with -O2, Intel Xeon with -O3, and AMD EPYC 9004 without compiler optimization flags. For each setting, we re-run the expert-crafted baselines and recalibrate the time and memory limits of subtasks accordingly before computing dual@10 with $\tau = \sigma = 1.2$.

As shown in Table 11, the absolute values of dual@10 vary slightly across different hardware and compiler configurations, but the model rankings remain identical in all four settings. This result suggests that, after per-environment recalibration, BEST provides stable comparative evaluations across different execution environments. The observation is also consistent with the design of dual@$k$: the metric is computed from subtask-level pass/fail outcomes under calibrated time and space

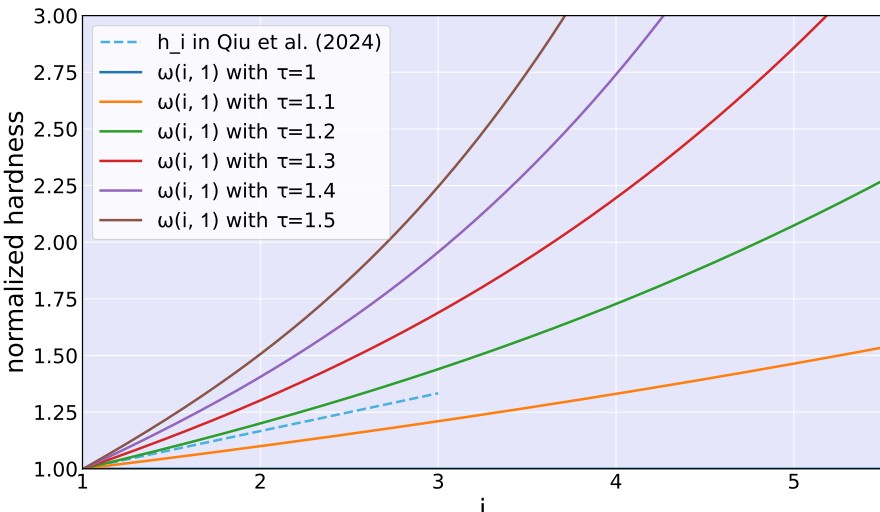

*Figure 5.* The normalized hardness $h_i$ and $\omega(i, 1)$ with different $\tau$. The curves plot the $\omega(i, 1)$ as a function of $i$. The dashed curve represents the values of $h_i/h_1$ at different $i$, where the values of $h_i$ are from the default setting in Qiu et al. (2024).

*Table 10.* Sensitivity of dual@10 to hyper-parameter $\alpha$ ($\tau = \sigma = \alpha$) for the top-10 LLMs. All ranks remain unchanged across $\alpha \in \{1.1, 1.2, 1.3, 1.4\}$.

| Model | $\alpha = 1.1$ | $\alpha = 1.2$ | $\alpha = 1.3$ | $\alpha = 1.4$ |
|---|---|---|---|---|
| Gemini-3-Pro-Preview | 0.852 | 0.849 | 0.847 | 0.844 |
| GPT-5.2 | 0.815 | 0.811 | 0.808 | 0.805 |
| Gemini-2.5-Pro | 0.769 | 0.764 | 0.760 | 0.756 |
| DeepSeek-V3.2-Thinking | 0.689 | 0.678 | 0.669 | 0.659 |
| DeepSeek-R1 | 0.644 | 0.635 | 0.627 | 0.620 |
| Gemini-2.5-Flash | 0.644 | 0.633 | 0.622 | 0.613 |
| DeepSeek-V3.2 | 0.640 | 0.630 | 0.621 | 0.612 |
| O4-Mini | 0.638 | 0.628 | 0.619 | 0.610 |
| QwQ-32B | 0.629 | 0.621 | 0.614 | 0.606 |
| Claude-Sonnet-4-5 | 0.606 | 0.596 | 0.588 | 0.580 |

limits, rather than directly from raw runtime or memory measurements of generated programs. Therefore, hardware and compiler differences mainly affect the calibration of subtask limits, while the resulting model-level comparison remains stable in this experiment.

### G.6. Efficiency among Different Difficulties

We evaluate the efficiency of LLMs among different difficulties in Table 12.

### G.7. Efficiency among Different Categorizations

We evaluate the efficiency of LLMs among different categorizations in Table 13-15. For time efficiency, Table 13 presents the values of dual@10 when $\tau = 1.2, \sigma = 0$ for LLMs. For space efficiency, Table 14 presents the values of dual@10 when $\tau = 0, \sigma = 1.2$. For time-space dual efficiency, Table 15 presents the values of dual@10 when $\tau = \sigma = 1.2$.

### G.8. Detailed Performance of Top 10 LLMs on BEST

We present the detailed performance of the TOP 10 LLMs that achieve the best time-space dual efficiency in Table 2. The winners are Gemini-3-Pro-Preview (Table 16), GPT-5.2 (Table 17), Gemini-2.5-Pro (Table 18), DeepSeek-V3.2-Thinking (Table 19), DeepSeek-R1 (Table 20), Gemini-2.5-Flash (Table 21), DeepSeek-V3.2 (Table 22), O4-Mini (Table 23),

*Table 11.* Cross-environment stability of dual@10 on the top-5 models across 440 tasks. Model rankings are identical across all four settings.

| Environment | GPT-5.2 | Gemini-3-Pro-Preview | Gemini-2.5-Pro | DeepSeek-V3.2-Thinking | DeepSeek-R1 |
|---|---|---|---|---|---|
| Intel Xeon, no flag | 0.811 | 0.849 | 0.764 | 0.678 | 0.635 |
| Intel Xeon, -O2 | 0.817 | 0.855 | 0.771 | 0.686 | 0.640 |
| Intel Xeon, -O3 | 0.817 | 0.855 | 0.772 | 0.687 | 0.641 |
| AMD, no flag | 0.817 | 0.855 | 0.768 | 0.684 | 0.639 |

QwQ-32B (Table 24), and Claude-Sonnet-4-5-20250929 (Table 25).

*Table 12.* Efficiency of codes generated by LLMs dual@10 among different levels of difficulty. The most efficient result for each difficulty is bolded and underlined.

| Model | Time Efficiency ($\tau = 1.2, \sigma = 0$) | | | | Space Efficiency ($\tau = 0, \sigma = 1.2$) | | | | Time-Space Dual Efficiency ($\tau = \sigma = 1.2$) | | | |
|---|---|---|---|---|---|---|---|---|---|---|---|---|
| | Easy | Medium | Hard | Overall | Easy | Medium | Hard | Overall | Easy | Medium | Hard | Overall |
| **Open-Source Models** | | | | | | | | | | | | |
| CodeLlama-34B-Hf | 0.345 | 0.088 | 0.012 | 0.108 | 0.188 | 0.054 | 0.010 | 0.063 | 0.158 | 0.033 | 0.004 | 0.046 |
| CodeLlama-34B-Instruct-Hf | 0.386 | 0.078 | 0.004 | 0.110 | 0.244 | 0.046 | 0.004 | 0.069 | 0.208 | 0.024 | 0.001 | 0.052 |
| CodeLlama-13B-Hf | 0.297 | 0.067 | 0.003 | 0.087 | 0.169 | 0.045 | 0.003 | 0.053 | 0.135 | 0.027 | 0.001 | 0.038 |
| CodeLlama-13B-Instruct-Hf | 0.319 | 0.074 | 0.011 | 0.097 | 0.190 | 0.041 | 0.008 | 0.057 | 0.155 | 0.026 | 0.004 | 0.043 |
| DeepSeek-V3.2-Thinking | 0.954 | **0.860** | **0.658** | **0.795** | 0.945 | **0.850** | **0.608** | **0.768** | 0.914 | **0.769** | **0.481** | **0.678** |
| DeepSeek-V3.2 | 0.920 | 0.849 | 0.543 | 0.736 | 0.939 | 0.837 | 0.501 | 0.718 | 0.878 | 0.747 | 0.404 | 0.630 |
| DeepSeek-R1 | 0.956 | 0.854 | 0.519 | 0.734 | 0.955 | 0.823 | 0.483 | 0.707 | **0.927** | 0.747 | 0.394 | 0.635 |
| DeepSeek-V3-0324 | 0.946 | 0.855 | 0.396 | 0.682 | 0.927 | 0.834 | 0.351 | 0.651 | 0.881 | 0.736 | 0.297 | 0.582 |
| DeepSeek-V3 | **0.973** | 0.827 | 0.331 | 0.650 | 0.960 | 0.822 | 0.291 | 0.629 | 0.895 | 0.714 | 0.218 | 0.544 |
| DeepSeek-Coder-V2-Lite-Instruct | 0.826 | 0.542 | 0.115 | 0.422 | 0.835 | 0.504 | 0.123 | 0.412 | 0.746 | 0.376 | 0.057 | 0.318 |
| DeepSeek-Coder-V2-Lite-Base | 0.896 | 0.581 | 0.097 | 0.443 | 0.890 | 0.535 | 0.084 | 0.419 | 0.807 | 0.384 | 0.041 | 0.327 |
| DeepSeek-Coder-33B-Instruct | 0.825 | 0.522 | 0.072 | 0.396 | 0.695 | 0.361 | 0.045 | 0.297 | 0.640 | 0.280 | 0.023 | 0.246 |
| DeepSeek-Coder-33B-Base | 0.777 | 0.473 | 0.062 | 0.363 | 0.635 | 0.304 | 0.036 | 0.259 | 0.585 | 0.237 | 0.018 | 0.216 |
| DeepSeek-Coder-6.7B-Instruct | 0.528 | 0.208 | 0.012 | 0.191 | 0.315 | 0.113 | 0.005 | 0.109 | 0.276 | 0.084 | 0.004 | 0.089 |
| DeepSeek-Coder-6.7B-Base | 0.464 | 0.167 | 0.009 | 0.162 | 0.222 | 0.076 | 0.003 | 0.075 | 0.205 | 0.059 | 0.003 | 0.065 |
| Qwen3-235B-A22B | **0.973** | 0.834 | 0.333 | 0.653 | **0.962** | 0.803 | 0.302 | 0.627 | 0.911 | 0.708 | 0.216 | 0.544 |
| Qwen3-Coder-30B-A3B-Instruct | 0.957 | 0.784 | 0.256 | 0.599 | 0.943 | 0.758 | 0.232 | 0.576 | 0.886 | 0.663 | 0.162 | 0.499 |
| QwQ-32B | 0.949 | 0.821 | 0.525 | 0.723 | 0.949 | 0.787 | 0.450 | 0.679 | 0.921 | 0.729 | 0.378 | 0.621 |
| Qwen2.5-Coder-32B | 0.965 | 0.802 | 0.196 | 0.583 | 0.930 | 0.748 | 0.146 | 0.534 | 0.886 | 0.636 | 0.089 | 0.459 |
| Qwen2.5-Coder-14B | 0.942 | 0.675 | 0.136 | 0.504 | 0.906 | 0.596 | 0.114 | 0.458 | 0.861 | 0.495 | 0.065 | 0.390 |
| Qwen2.5-Coder-7B | 0.877 | 0.568 | 0.063 | 0.420 | 0.826 | 0.479 | 0.052 | 0.371 | 0.770 | 0.377 | 0.025 | 0.310 |
| Yi-34B | 0.663 | 0.237 | 0.025 | 0.235 | 0.525 | 0.146 | 0.023 | 0.171 | 0.465 | 0.095 | 0.009 | 0.134 |
| Yi-9B | 0.647 | 0.243 | 0.008 | 0.227 | 0.417 | 0.134 | 0.010 | 0.140 | 0.363 | 0.097 | 0.003 | 0.112 |
| Llama-3.1-70B | 0.865 | 0.612 | 0.172 | 0.480 | 0.876 | 0.577 | 0.175 | 0.470 | 0.807 | 0.490 | 0.143 | 0.410 |
| Llama-3.1-8B | 0.578 | 0.302 | 0.028 | 0.244 | 0.603 | 0.300 | 0.026 | 0.248 | 0.524 | 0.214 | 0.015 | 0.194 |
| Mixtral-8x7B-V0.1 | 0.567 | 0.220 | 0.012 | 0.203 | 0.447 | 0.139 | 0.014 | 0.149 | 0.378 | 0.091 | 0.004 | 0.113 |
| Mixtral-8x7B-Instruct-V0.1 | 0.521 | 0.208 | 0.009 | 0.188 | 0.447 | 0.145 | 0.012 | 0.151 | 0.382 | 0.091 | 0.004 | 0.114 |
| StarCoder2-15B | 0.170 | 0.043 | 0.005 | 0.053 | 0.056 | 0.017 | 0.001 | 0.019 | 0.053 | 0.012 | 0.001 | 0.016 |
| StarCoder2-15B-Instruct-V0.1 | 0.563 | 0.184 | 0.025 | 0.195 | 0.339 | 0.104 | 0.015 | 0.115 | 0.292 | 0.064 | 0.007 | 0.087 |
| Mistral-7B-Instruct-V0.3 | 0.347 | 0.093 | 0.007 | 0.109 | 0.269 | 0.062 | 0.006 | 0.080 | 0.209 | 0.035 | 0.002 | 0.057 |
| CodeGemma-7B | 0.580 | 0.188 | 0.021 | 0.198 | 0.463 | 0.136 | 0.020 | 0.154 | 0.396 | 0.088 | 0.008 | 0.118 |
| WizardCoder-33B-V1.1 | 0.837 | 0.483 | 0.041 | 0.370 | 0.684 | 0.311 | 0.028 | 0.269 | 0.615 | 0.237 | 0.013 | 0.220 |
| Phind-CodeLlama-34B-V1 | 0.634 | 0.185 | 0.027 | 0.210 | 0.428 | 0.119 | 0.016 | 0.139 | 0.371 | 0.076 | 0.007 | 0.107 |
| Magicoder-S-CL-7B | 0.267 | 0.044 | 0.013 | 0.076 | 0.133 | 0.028 | 0.009 | 0.041 | 0.118 | 0.015 | 0.004 | 0.031 |
| StableLM-2-12B | 0.378 | 0.118 | 0.015 | 0.128 | 0.278 | 0.096 | 0.015 | 0.099 | 0.219 | 0.057 | 0.006 | 0.068 |
| XwinCoder-34B | 0.415 | 0.093 | 0.001 | 0.120 | 0.232 | 0.054 | 0.001 | 0.068 | 0.201 | 0.032 | 0.000 | 0.053 |
| **Closed-Source Models** | | | | | | | | | | | | |
| Gemini-3-Pro-Preview | 0.953 | 0.897 | **0.854** | **0.890** | 0.955 | 0.880 | **0.815** | **0.868** | **0.940** | **0.870** | **0.787** | **0.849** |
| Gemini-2.5-Pro | 0.944 | **0.900** | 0.693 | 0.825 | 0.926 | **0.883** | 0.648 | 0.796 | 0.904 | 0.867 | 0.597 | 0.764 |
| Gemini-2.5-Flash | 0.935 | 0.878 | 0.608 | 0.777 | 0.695 | 0.805 | 0.536 | 0.671 | 0.678 | 0.780 | 0.477 | 0.633 |
| GPT-5.2 | 0.931 | 0.888 | 0.788 | 0.855 | 0.941 | 0.875 | 0.763 | 0.841 | 0.926 | 0.853 | 0.717 | 0.811 |
| O4-Mini | 0.951 | 0.887 | 0.564 | 0.765 | 0.945 | 0.829 | 0.443 | 0.692 | 0.911 | 0.755 | 0.375 | 0.628 |
| GPT-4.1 | 0.948 | 0.782 | 0.383 | 0.650 | 0.940 | 0.768 | 0.345 | 0.627 | 0.888 | 0.670 | 0.275 | 0.549 |
| GPT-4o | **0.986** | 0.828 | 0.263 | 0.625 | 0.978 | 0.810 | 0.199 | 0.590 | 0.923 | 0.703 | 0.156 | 0.519 |
| GPT-4o-Mini | 0.942 | 0.668 | 0.122 | 0.496 | 0.920 | 0.643 | 0.106 | 0.476 | 0.858 | 0.525 | 0.058 | 0.398 |
| GPT-3.5-Turbo-1106 | 0.970 | 0.754 | 0.146 | 0.544 | 0.927 | 0.640 | 0.099 | 0.473 | 0.866 | 0.528 | 0.064 | 0.403 |
| Grok-3 | 0.951 | 0.771 | 0.307 | 0.614 | 0.962 | 0.783 | 0.293 | 0.615 | 0.890 | 0.655 | 0.208 | 0.516 |
| Grok-3-Mini | 0.955 | 0.749 | 0.254 | 0.584 | 0.943 | 0.689 | 0.195 | 0.534 | 0.907 | 0.634 | 0.154 | 0.489 |
| Claude-Sonnet-4-5-20250929 | 0.963 | 0.858 | 0.389 | 0.684 | **0.982** | 0.844 | 0.423 | 0.697 | 0.927 | 0.741 | 0.303 | 0.596 |
| Claude-3-7-Sonnet-20250219 | 0.953 | 0.795 | 0.341 | 0.638 | 0.971 | 0.796 | 0.348 | 0.645 | 0.882 | 0.669 | 0.230 | 0.529 |
| Claude-3-5-Sonnet-20241022 | 0.978 | 0.811 | 0.291 | 0.628 | 0.963 | 0.791 | 0.263 | 0.606 | 0.902 | 0.683 | 0.185 | 0.520 |
| Claude-3-Haiku-20240307 | 0.892 | 0.694 | 0.240 | 0.546 | 0.905 | 0.725 | 0.207 | 0.547 | 0.822 | 0.576 | 0.140 | 0.445 |

*Table 13.* Time efficiency of codes generated by LLMs dual@10 ($\tau = 1.2, \sigma = 0$) among different types of algorithm. The most efficient result for each categorization is bolded and underlined.

| Model | String | Sort | Greedy | Graph | Data Structure | Two Pointers | Math | Search | DP | Other | Overall |
|---|---|---|---|---|---|---|---|---|---|---|---|
| | | | | | **Open-Source Models** | | | | | | |
| CodeLlama-34B-Hf | 0.165 | 0.101 | 0.060 | 0.006 | 0.070 | 0.099 | 0.144 | 0.084 | 0.051 | 0.119 | 0.108 |
| CodeLlama-34B-Instruct-Hf | 0.171 | 0.092 | 0.068 | 0.027 | 0.072 | 0.102 | 0.133 | 0.102 | 0.029 | 0.107 | 0.110 |
| CodeLlama-13B-Hf | 0.164 | 0.063 | 0.059 | 0.023 | 0.042 | 0.076 | 0.098 | 0.082 | 0.024 | 0.104 | 0.087 |
| CodeLlama-13B-Instruct-Hf | 0.162 | 0.082 | 0.052 | 0.025 | 0.051 | 0.064 | 0.115 | 0.066 | 0.017 | 0.118 | 0.097 |
| DeepSeek-V3.2-Thinking | **_0.857_** | **_0.870_** | **_0.865_** | **_0.794_** | **_0.671_** | **_0.933_** | **_0.793_** | 0.781 | **_0.755_** | **_0.829_** | **_0.795_** |
| DeepSeek-V3.2 | 0.782 | 0.818 | 0.814 | 0.744 | 0.596 | 0.876 | 0.739 | **_0.819_** | 0.652 | 0.761 | 0.736 |
| DeepSeek-R1 | 0.794 | 0.797 | 0.791 | 0.715 | 0.548 | 0.875 | 0.721 | 0.771 | 0.639 | 0.819 | 0.734 |
| DeepSeek-V3-0324 | 0.757 | 0.759 | 0.686 | 0.612 | 0.490 | 0.812 | 0.661 | 0.664 | 0.546 | 0.775 | 0.682 |
| DeepSeek-V3 | 0.750 | 0.672 | 0.654 | 0.622 | 0.494 | 0.704 | 0.667 | 0.662 | 0.492 | 0.696 | 0.650 |
| DeepSeek-Coder-V2-Lite-Instruct | 0.560 | 0.403 | 0.337 | 0.453 | 0.323 | 0.366 | 0.403 | 0.430 | 0.327 | 0.505 | 0.422 |
| DeepSeek-Coder-V2-Lite-Base | 0.566 | 0.399 | 0.372 | 0.400 | 0.396 | 0.375 | 0.410 | 0.424 | 0.317 | 0.443 | 0.443 |
| DeepSeek-Coder-33B-Instruct | 0.563 | 0.347 | 0.282 | 0.363 | 0.239 | 0.323 | 0.360 | 0.309 | 0.240 | 0.482 | 0.396 |
| DeepSeek-Coder-33B-Base | 0.527 | 0.315 | 0.245 | 0.334 | 0.212 | 0.279 | 0.336 | 0.270 | 0.216 | 0.444 | 0.363 |
| DeepSeek-Coder-6.7B-Instruct | 0.295 | 0.162 | 0.141 | 0.113 | 0.080 | 0.158 | 0.159 | 0.134 | 0.100 | 0.210 | 0.191 |
| DeepSeek-Coder-6.7B-Base | 0.259 | 0.141 | 0.122 | 0.079 | 0.063 | 0.139 | 0.134 | 0.110 | 0.081 | 0.164 | 0.162 |
| Qwen3-235B-A22B | 0.765 | 0.655 | 0.627 | 0.653 | 0.503 | 0.709 | 0.654 | 0.699 | 0.490 | 0.770 | 0.653 |
| Qwen3-Coder-30B-A3B-Instruct | 0.764 | 0.544 | 0.508 | 0.572 | 0.428 | 0.594 | 0.592 | 0.601 | 0.416 | 0.745 | 0.599 |
| QwQ-32B | 0.751 | 0.806 | 0.785 | 0.702 | 0.613 | 0.855 | 0.720 | 0.774 | 0.642 | 0.792 | 0.723 |
| Qwen2.5-Coder-32B | 0.752 | 0.601 | 0.533 | 0.494 | 0.420 | 0.606 | 0.568 | 0.547 | 0.410 | 0.664 | 0.583 |
| Qwen2.5-Coder-14B | 0.676 | 0.459 | 0.431 | 0.458 | 0.359 | 0.440 | 0.489 | 0.462 | 0.317 | 0.585 | 0.504 |
| Qwen2.5-Coder-7B | 0.606 | 0.359 | 0.300 | 0.290 | 0.269 | 0.331 | 0.390 | 0.390 | 0.266 | 0.556 | 0.420 |
| Yi-34B | 0.335 | 0.222 | 0.154 | 0.166 | 0.139 | 0.171 | 0.246 | 0.210 | 0.169 | 0.193 | 0.235 |
| Yi-9B | 0.338 | 0.232 | 0.173 | 0.159 | 0.112 | 0.168 | 0.216 | 0.204 | 0.112 | 0.237 | 0.227 |
| Llama-3.1-70B | 0.513 | 0.476 | 0.504 | 0.400 | 0.383 | 0.475 | 0.511 | 0.434 | 0.441 | 0.383 | 0.480 |
| Llama-3.1-8B | 0.355 | 0.218 | 0.244 | 0.120 | 0.147 | 0.253 | 0.272 | 0.238 | 0.160 | 0.281 | 0.244 |
| Mixtral-8x7B-V0.1 | 0.262 | 0.210 | 0.133 | 0.141 | 0.150 | 0.155 | 0.219 | 0.163 | 0.086 | 0.250 | 0.203 |
| Mixtral-8x7B-Instruct-V0.1 | 0.257 | 0.171 | 0.102 | 0.176 | 0.131 | 0.119 | 0.239 | 0.145 | 0.055 | 0.220 | 0.188 |
| StarCoder2-15B | 0.074 | 0.075 | 0.036 | 0.000 | 0.000 | 0.057 | 0.045 | 0.038 | 0.002 | 0.084 | 0.053 |
| StarCoder2-15B-Instruct-V0.1 | 0.301 | 0.180 | 0.122 | 0.156 | 0.128 | 0.182 | 0.177 | 0.184 | 0.107 | 0.216 | 0.195 |
| Mistral-7B-Instruct-V0.3 | 0.178 | 0.110 | 0.071 | 0.054 | 0.070 | 0.114 | 0.087 | 0.124 | 0.026 | 0.062 | 0.109 |
| CodeGemma-7B | 0.296 | 0.178 | 0.162 | 0.120 | 0.074 | 0.150 | 0.209 | 0.141 | 0.094 | 0.202 | 0.198 |
| WizardCoder-33B-V1.1 | 0.556 | 0.342 | 0.288 | 0.359 | 0.242 | 0.287 | 0.326 | 0.310 | 0.207 | 0.383 | 0.370 |
| Phind-CodeLlama-34B-V1 | 0.326 | 0.245 | 0.143 | 0.048 | 0.154 | 0.186 | 0.249 | 0.147 | 0.068 | 0.239 | 0.210 |
| Magicoder-S-CL-7B | 0.134 | 0.044 | 0.055 | 0.000 | 0.020 | 0.091 | 0.083 | 0.037 | 0.021 | 0.110 | 0.076 |
| StableLM-2-12B | 0.227 | 0.128 | 0.086 | 0.032 | 0.090 | 0.149 | 0.120 | 0.095 | 0.043 | 0.138 | 0.128 |
| XwinCoder-34B | 0.241 | 0.127 | 0.081 | 0.000 | 0.032 | 0.087 | 0.126 | 0.077 | 0.026 | 0.110 | 0.120 |
| | | | | | **Closed-Source Models** | | | | | | |
| Gemini-3-Pro-Preview | **_0.929_** | **_0.923_** | **_0.951_** | **_0.900_** | **_0.863_** | **_0.965_** | **_0.888_** | **_0.911_** | **_0.868_** | 0.852 | **_0.890_** |
| Gemini-2.5-Pro | 0.911 | 0.876 | 0.883 | 0.782 | 0.756 | 0.926 | 0.819 | 0.830 | 0.772 | 0.812 | 0.825 |
| Gemini-2.5-Flash | 0.900 | 0.882 | 0.847 | 0.680 | 0.650 | 0.934 | 0.770 | 0.755 | 0.688 | 0.726 | 0.777 |
| GPT-5.2 | 0.861 | 0.903 | 0.897 | 0.821 | 0.833 | 0.955 | 0.865 | 0.895 | 0.831 | **_0.870_** | 0.855 |
| O4-Mini | 0.862 | 0.836 | 0.840 | 0.706 | 0.648 | 0.904 | 0.746 | 0.813 | 0.663 | 0.800 | 0.765 |
| GPT-4.1 | 0.798 | 0.627 | 0.677 | 0.684 | 0.480 | 0.746 | 0.650 | 0.686 | 0.546 | 0.705 | 0.650 |
| GPT-4o | 0.721 | 0.596 | 0.603 | 0.608 | 0.462 | 0.685 | 0.643 | 0.581 | 0.469 | 0.656 | 0.625 |
| GPT-4o-Mini | 0.654 | 0.475 | 0.428 | 0.390 | 0.342 | 0.468 | 0.486 | 0.527 | 0.354 | 0.640 | 0.496 |
| GPT-3.5-Turbo-1106 | 0.692 | 0.496 | 0.467 | 0.438 | 0.411 | 0.593 | 0.538 | 0.546 | 0.396 | 0.613 | 0.544 |
| Grok-3 | 0.731 | 0.660 | 0.618 | 0.534 | 0.470 | 0.686 | 0.601 | 0.655 | 0.464 | 0.567 | 0.614 |
| Grok-3-Mini | 0.685 | 0.664 | 0.570 | 0.566 | 0.427 | 0.572 | 0.566 | 0.518 | 0.405 | 0.584 | 0.584 |
| Claude-Sonnet-4-5-20250929 | 0.794 | 0.695 | 0.685 | 0.617 | 0.529 | 0.731 | 0.679 | 0.685 | 0.542 | 0.750 | 0.684 |
| Claude-3-7-Sonnet-20250219 | 0.748 | 0.643 | 0.664 | 0.558 | 0.515 | 0.705 | 0.610 | 0.667 | 0.499 | 0.633 | 0.638 |
| Claude-3-5-Sonnet-20241022 | 0.733 | 0.582 | 0.591 | 0.613 | 0.490 | 0.655 | 0.643 | 0.604 | 0.483 | 0.656 | 0.628 |
| Claude-3-Haiku-20240307 | 0.686 | 0.464 | 0.478 | 0.584 | 0.444 | 0.482 | 0.504 | 0.501 | 0.431 | 0.658 | 0.546 |

*Table 14.* Space efficiency of codes generated by LLMs dual@10 ($\tau = 0, \sigma = 1.2$) among different types of algorithm. The most efficient result for each categorization is bolded and underlined.

| Model | String | Sort | Greedy | Graph | Data Structure | Two Pointers | Math | Search | DP | Other | Overall |
|---|---|---|---|---|---|---|---|---|---|---|---|
| | | | | | *Open-Source Models* | | | | | | |
| CodeLlama-34B-Hf | 0.080 | 0.073 | 0.039 | 0.006 | 0.052 | 0.080 | 0.067 | 0.062 | 0.024 | 0.082 | 0.063 |
| CodeLlama-34B-Instruct-Hf | 0.097 | 0.074 | 0.051 | 0.012 | 0.056 | 0.076 | 0.074 | 0.068 | 0.017 | 0.062 | 0.069 |
| CodeLlama-13B-Hf | 0.072 | 0.054 | 0.044 | 0.014 | 0.035 | 0.058 | 0.047 | 0.044 | 0.015 | 0.063 | 0.053 |
| CodeLlama-13B-Instruct-Hf | 0.069 | 0.063 | 0.043 | 0.012 | 0.031 | 0.046 | 0.062 | 0.034 | 0.016 | 0.058 | 0.057 |
| DeepSeek-V3.2-Thinking | **0.826** | **0.822** | **0.809** | **0.800** | **0.650** | **0.859** | **0.754** | 0.736 | **0.717** | **0.824** | **0.768** |
| DeepSeek-V3.2 | 0.796 | 0.772 | 0.742 | 0.767 | 0.604 | 0.812 | 0.718 | **0.761** | 0.628 | 0.776 | 0.718 |
| DeepSeek-R1 | 0.781 | 0.781 | 0.737 | 0.714 | 0.501 | 0.815 | 0.684 | 0.723 | 0.628 | 0.758 | 0.707 |
| DeepSeek-V3-0324 | 0.734 | 0.699 | 0.614 | 0.604 | 0.430 | 0.793 | 0.639 | 0.638 | 0.537 | 0.734 | 0.651 |
| DeepSeek-V3 | 0.748 | 0.609 | 0.611 | 0.576 | 0.458 | 0.674 | 0.646 | 0.621 | 0.463 | 0.660 | 0.629 |
| DeepSeek-Coder-V2-Lite-Instruct | 0.502 | 0.325 | 0.271 | 0.426 | 0.355 | 0.326 | 0.433 | 0.364 | 0.329 | 0.532 | 0.412 |
| DeepSeek-Coder-V2-Lite-Base | 0.536 | 0.322 | 0.319 | 0.368 | 0.397 | 0.350 | 0.399 | 0.369 | 0.291 | 0.409 | 0.419 |
| DeepSeek-Coder-33B-Instruct | 0.419 | 0.241 | 0.190 | 0.307 | 0.176 | 0.224 | 0.289 | 0.225 | 0.179 | 0.334 | 0.297 |
| DeepSeek-Coder-33B-Base | 0.381 | 0.211 | 0.161 | 0.259 | 0.155 | 0.187 | 0.263 | 0.185 | 0.152 | 0.270 | 0.259 |
| DeepSeek-Coder-6.7B-Instruct | 0.143 | 0.091 | 0.081 | 0.072 | 0.046 | 0.089 | 0.102 | 0.091 | 0.049 | 0.115 | 0.109 |
| DeepSeek-Coder-6.7B-Base | 0.093 | 0.065 | 0.062 | 0.035 | 0.035 | 0.070 | 0.074 | 0.067 | 0.027 | 0.071 | 0.075 |
| Qwen3-235B-A22B | 0.762 | 0.627 | 0.572 | 0.618 | 0.498 | 0.675 | 0.636 | 0.652 | 0.455 | 0.698 | 0.627 |
| Qwen3-Coder-30B-A3B-Instruct | 0.753 | 0.475 | 0.460 | 0.531 | 0.496 | 0.560 | 0.557 | 0.504 | 0.384 | 0.691 | 0.576 |
| QwQ-32B | 0.735 | 0.743 | 0.700 | 0.687 | 0.529 | 0.753 | 0.658 | 0.702 | 0.592 | 0.720 | 0.679 |
| Qwen2.5-Coder-32B | 0.700 | 0.508 | 0.448 | 0.492 | 0.397 | 0.519 | 0.504 | 0.491 | 0.373 | 0.608 | 0.534 |
| Qwen2.5-Coder-14B | 0.672 | 0.407 | 0.373 | 0.404 | 0.340 | 0.381 | 0.452 | 0.420 | 0.295 | 0.459 | 0.458 |
| Qwen2.5-Coder-7B | 0.539 | 0.327 | 0.274 | 0.254 | 0.265 | 0.255 | 0.355 | 0.280 | 0.221 | 0.419 | 0.371 |
| Yi-34B | 0.286 | 0.152 | 0.116 | 0.078 | 0.102 | 0.120 | 0.180 | 0.141 | 0.128 | 0.140 | 0.171 |
| Yi-9B | 0.257 | 0.120 | 0.086 | 0.064 | 0.069 | 0.113 | 0.137 | 0.103 | 0.075 | 0.158 | 0.140 |
| Llama-3.1-70B | 0.524 | 0.470 | 0.486 | 0.423 | 0.365 | 0.440 | 0.511 | 0.392 | 0.427 | 0.348 | 0.470 |
| Llama-3.1-8B | 0.397 | 0.246 | 0.258 | 0.141 | 0.140 | 0.243 | 0.253 | 0.260 | 0.157 | 0.288 | 0.248 |
| Mixtral-8x7B-V0.1 | 0.201 | 0.179 | 0.089 | 0.075 | 0.125 | 0.158 | 0.142 | 0.149 | 0.056 | 0.194 | 0.149 |
| Mixtral-8x7B-Instruct-V0.1 | 0.202 | 0.157 | 0.089 | 0.117 | 0.130 | 0.127 | 0.162 | 0.135 | 0.054 | 0.168 | 0.151 |
| StarCoder2-15B | 0.027 | 0.022 | 0.012 | 0.000 | 0.000 | 0.024 | 0.018 | 0.013 | 0.002 | 0.025 | 0.019 |
| StarCoder2-15B-Instruct-V0.1 | 0.184 | 0.096 | 0.062 | 0.075 | 0.075 | 0.105 | 0.105 | 0.112 | 0.067 | 0.163 | 0.115 |
| Mistral-7B-Instruct-V0.3 | 0.108 | 0.090 | 0.059 | 0.025 | 0.056 | 0.085 | 0.059 | 0.066 | 0.025 | 0.062 | 0.080 |
| CodeGemma-7B | 0.235 | 0.163 | 0.119 | 0.114 | 0.071 | 0.139 | 0.150 | 0.136 | 0.064 | 0.155 | 0.154 |
| WizardCoder-33B-V1.1 | 0.418 | 0.239 | 0.172 | 0.261 | 0.172 | 0.188 | 0.243 | 0.220 | 0.164 | 0.295 | 0.269 |
| Phind-CodeLlama-34B-V1 | 0.224 | 0.123 | 0.082 | 0.047 | 0.102 | 0.135 | 0.167 | 0.113 | 0.060 | 0.141 | 0.139 |
| Magicoder-S-CL-7B | 0.060 | 0.031 | 0.037 | 0.000 | 0.012 | 0.042 | 0.047 | 0.027 | 0.018 | 0.058 | 0.041 |
| StableLM-2-12B | 0.145 | 0.123 | 0.085 | 0.027 | 0.066 | 0.121 | 0.072 | 0.070 | 0.028 | 0.116 | 0.099 |
| XwinCoder-34B | 0.140 | 0.063 | 0.039 | 0.000 | 0.017 | 0.064 | 0.063 | 0.053 | 0.020 | 0.095 | 0.068 |
| | | | | | *Closed-Source Models* | | | | | | |
| Gemini-3-Pro-Preview | **0.916** | **0.911** | **0.935** | **0.896** | 0.800 | 0.937 | **0.862** | **0.881** | **0.849** | 0.826 | **0.868** |
| Gemini-2.5-Pro | 0.865 | 0.858 | 0.853 | 0.815 | 0.687 | 0.914 | 0.784 | 0.794 | 0.735 | 0.788 | 0.796 |
| Gemini-2.5-Flash | 0.733 | 0.726 | 0.719 | 0.622 | 0.579 | 0.812 | 0.676 | 0.656 | 0.643 | 0.653 | 0.671 |
| GPT-5.2 | 0.863 | 0.904 | 0.883 | 0.834 | **0.812** | **0.963** | 0.851 | 0.871 | 0.804 | **0.830** | 0.841 |
| O4-Mini | 0.794 | 0.710 | 0.693 | 0.680 | 0.528 | 0.801 | 0.691 | 0.708 | 0.560 | 0.751 | 0.692 |
| GPT-4.1 | 0.780 | 0.583 | 0.609 | 0.701 | 0.421 | 0.731 | 0.642 | 0.677 | 0.516 | 0.683 | 0.627 |
| GPT-4o | 0.689 | 0.548 | 0.535 | 0.577 | 0.455 | 0.654 | 0.590 | 0.562 | 0.414 | 0.577 | 0.590 |
| GPT-4o-Mini | 0.657 | 0.393 | 0.385 | 0.366 | 0.362 | 0.443 | 0.467 | 0.419 | 0.340 | 0.561 | 0.476 |
| GPT-3.5-Turbo-1106 | 0.641 | 0.395 | 0.380 | 0.344 | 0.367 | 0.487 | 0.482 | 0.429 | 0.343 | 0.491 | 0.473 |
| Grok-3 | 0.746 | 0.590 | 0.589 | 0.528 | 0.516 | 0.647 | 0.591 | 0.590 | 0.467 | 0.522 | 0.615 |
| Grok-3-Mini | 0.635 | 0.545 | 0.494 | 0.514 | 0.359 | 0.503 | 0.515 | 0.420 | 0.374 | 0.494 | 0.534 |
| Claude-Sonnet-4-5-20250929 | 0.807 | 0.676 | 0.686 | 0.624 | 0.611 | 0.710 | 0.692 | 0.666 | 0.576 | 0.704 | 0.697 |
| Claude-3-7-Sonnet-20250219 | 0.745 | 0.586 | 0.657 | 0.542 | 0.525 | 0.695 | 0.628 | 0.616 | 0.522 | 0.597 | 0.645 |
| Claude-3-5-Sonnet-20241022 | 0.697 | 0.522 | 0.533 | 0.583 | 0.493 | 0.602 | 0.594 | 0.520 | 0.481 | 0.589 | 0.606 |
| Claude-3-Haiku-20240307 | 0.627 | 0.454 | 0.463 | 0.563 | 0.469 | 0.405 | 0.535 | 0.461 | 0.461 | 0.547 | 0.547 |

*Table 15.* Time-space dual efficiency of codes generated by LLMs dual@10 ($\sigma = \tau = 1.2$) among different types of algorithm. The most efficient result for each categorization is bolded and underlined.

| Model | String | Sort | Greedy | Graph | Data Structure | Two Pointers | Math | Search | DP | Other | Overall |
|---|---|---|---|---|---|---|---|---|---|---|---|
| | | | | | **Open-Source Models** | | | | | | |
| CodeLlama-34B-Hf | 0.068 | 0.041 | 0.023 | 0.002 | 0.034 | 0.047 | 0.059 | 0.036 | 0.014 | 0.062 | 0.046 |
| CodeLlama-34B-Instruct-Hf | 0.073 | 0.043 | 0.034 | 0.007 | 0.038 | 0.056 | 0.064 | 0.056 | 0.011 | 0.040 | 0.052 |
| CodeLlama-13B-Hf | 0.057 | 0.032 | 0.028 | 0.006 | 0.021 | 0.034 | 0.045 | 0.033 | 0.007 | 0.045 | 0.038 |
| CodeLlama-13B-Instruct-Hf | 0.059 | 0.041 | 0.024 | 0.007 | 0.019 | 0.034 | 0.053 | 0.032 | 0.006 | 0.044 | 0.043 |
| DeepSeek-V3.2-Thinking | **0.762** | **0.710** | **0.727** | **0.664** | **0.514** | **0.808** | **0.680** | 0.674 | **0.610** | **0.730** | **0.678** |
| DeepSeek-V3.2 | 0.720 | 0.673 | 0.655 | 0.651 | 0.463 | 0.753 | 0.643 | **0.690** | 0.529 | 0.679 | 0.630 |
| DeepSeek-R1 | 0.714 | 0.651 | 0.648 | 0.610 | 0.430 | 0.726 | 0.622 | 0.650 | 0.550 | 0.690 | 0.635 |
| DeepSeek-V3-0324 | 0.672 | 0.605 | 0.564 | 0.524 | 0.371 | 0.716 | 0.574 | 0.559 | 0.476 | 0.650 | 0.582 |
| DeepSeek-V3 | 0.651 | 0.528 | 0.531 | 0.477 | 0.390 | 0.587 | 0.552 | 0.524 | 0.388 | 0.572 | 0.544 |
| DeepSeek-Coder-V2-Lite-Instruct | 0.434 | 0.263 | 0.212 | 0.323 | 0.244 | 0.236 | 0.321 | 0.289 | 0.239 | 0.398 | 0.318 |
| DeepSeek-Coder-V2-Lite-Base | 0.452 | 0.263 | 0.239 | 0.267 | 0.290 | 0.240 | 0.319 | 0.269 | 0.219 | 0.324 | 0.327 |
| DeepSeek-Coder-33B-Instruct | 0.374 | 0.208 | 0.159 | 0.214 | 0.133 | 0.184 | 0.229 | 0.171 | 0.133 | 0.271 | 0.246 |
| DeepSeek-Coder-33B-Base | 0.341 | 0.183 | 0.136 | 0.180 | 0.117 | 0.154 | 0.209 | 0.137 | 0.111 | 0.216 | 0.216 |
| DeepSeek-Coder-6.7B-Instruct | 0.126 | 0.075 | 0.061 | 0.051 | 0.037 | 0.075 | 0.085 | 0.072 | 0.039 | 0.093 | 0.089 |
| DeepSeek-Coder-6.7B-Base | 0.087 | 0.056 | 0.047 | 0.029 | 0.029 | 0.061 | 0.064 | 0.056 | 0.025 | 0.060 | 0.065 |
| Qwen3-235B-A22B | 0.693 | 0.516 | 0.489 | 0.507 | 0.394 | 0.593 | 0.552 | 0.547 | 0.391 | 0.625 | 0.544 |
| Qwen3-Coder-30B-A3B-Instruct | 0.669 | 0.404 | 0.396 | 0.444 | 0.372 | 0.467 | 0.502 | 0.450 | 0.323 | 0.628 | 0.499 |
| QwQ-32B | 0.678 | 0.666 | 0.639 | 0.559 | 0.444 | 0.697 | 0.608 | 0.646 | 0.545 | 0.684 | 0.621 |
| Qwen2.5-Coder-32B | 0.617 | 0.445 | 0.378 | 0.387 | 0.307 | 0.451 | 0.427 | 0.420 | 0.305 | 0.554 | 0.459 |
| Qwen2.5-Coder-14B | 0.584 | 0.363 | 0.315 | 0.321 | 0.264 | 0.312 | 0.376 | 0.340 | 0.212 | 0.404 | 0.390 |
| Qwen2.5-Coder-7B | 0.473 | 0.272 | 0.212 | 0.195 | 0.189 | 0.205 | 0.293 | 0.209 | 0.180 | 0.376 | 0.310 |
| Yi-34B | 0.247 | 0.126 | 0.086 | 0.050 | 0.064 | 0.092 | 0.148 | 0.098 | 0.078 | 0.095 | 0.134 |
| Yi-9B | 0.222 | 0.096 | 0.066 | 0.051 | 0.045 | 0.092 | 0.107 | 0.086 | 0.058 | 0.125 | 0.112 |
| Llama-3.1-70B | 0.427 | 0.397 | 0.432 | 0.352 | 0.316 | 0.422 | 0.446 | 0.353 | 0.377 | 0.297 | 0.410 |
| Llama-3.1-8B | 0.333 | 0.167 | 0.197 | 0.079 | 0.090 | 0.201 | 0.211 | 0.183 | 0.123 | 0.217 | 0.194 |
| Mixtral-8x7B-V0.1 | 0.169 | 0.122 | 0.060 | 0.053 | 0.083 | 0.093 | 0.122 | 0.099 | 0.030 | 0.148 | 0.113 |
| Mixtral-8x7B-Instruct-V0.1 | 0.170 | 0.114 | 0.059 | 0.081 | 0.080 | 0.081 | 0.139 | 0.100 | 0.029 | 0.119 | 0.114 |
| StarCoder2-15B | 0.024 | 0.021 | 0.010 | 0.000 | 0.000 | 0.019 | 0.015 | 0.010 | 0.001 | 0.023 | 0.016 |
| StarCoder2-15B-Instruct-V0.1 | 0.155 | 0.073 | 0.039 | 0.047 | 0.046 | 0.083 | 0.086 | 0.074 | 0.039 | 0.106 | 0.087 |
| Mistral-7B-Instruct-V0.3 | 0.087 | 0.057 | 0.036 | 0.015 | 0.034 | 0.052 | 0.047 | 0.050 | 0.009 | 0.033 | 0.057 |
| CodeGemma-7B | 0.203 | 0.114 | 0.090 | 0.059 | 0.037 | 0.097 | 0.122 | 0.093 | 0.044 | 0.123 | 0.118 |
| WizardCoder-33B-V1.1 | 0.361 | 0.196 | 0.142 | 0.185 | 0.130 | 0.153 | 0.199 | 0.166 | 0.119 | 0.243 | 0.220 |
| Phind-CodeLlama-34B-V1 | 0.189 | 0.102 | 0.057 | 0.021 | 0.064 | 0.107 | 0.140 | 0.081 | 0.032 | 0.116 | 0.107 |
| Magicoder-S-CL-7B | 0.052 | 0.016 | 0.022 | 0.000 | 0.006 | 0.027 | 0.039 | 0.013 | 0.009 | 0.037 | 0.031 |
| StableLM-2-12B | 0.124 | 0.076 | 0.048 | 0.019 | 0.040 | 0.078 | 0.052 | 0.046 | 0.013 | 0.073 | 0.068 |
| XwinCoder-34B | 0.116 | 0.053 | 0.032 | 0.000 | 0.010 | 0.052 | 0.049 | 0.049 | 0.009 | 0.051 | 0.053 |
| | | | | | **Closed-Source Models** | | | | | | |
| Gemini-3-Pro-Preview | **0.886** | **0.910** | **0.922** | **0.830** | **0.797** | 0.928 | **0.841** | **0.875** | **0.816** | 0.795 | **0.849** |
| Gemini-2.5-Pro | 0.839 | 0.856 | 0.834 | 0.717 | 0.662 | 0.893 | 0.752 | 0.793 | 0.688 | 0.771 | 0.764 |
| Gemini-2.5-Flash | 0.700 | 0.711 | 0.684 | 0.544 | 0.529 | 0.786 | 0.649 | 0.642 | 0.584 | 0.625 | 0.633 |
| GPT-5.2 | 0.822 | 0.894 | 0.872 | 0.763 | 0.768 | **0.933** | 0.819 | 0.864 | 0.758 | **0.799** | 0.811 |
| O4-Mini | 0.740 | 0.634 | 0.631 | 0.556 | 0.466 | 0.718 | 0.644 | 0.640 | 0.496 | 0.676 | 0.628 |
| GPT-4.1 | 0.707 | 0.499 | 0.542 | 0.571 | 0.348 | 0.655 | 0.571 | 0.579 | 0.440 | 0.581 | 0.549 |
| GPT-4o | 0.626 | 0.482 | 0.469 | 0.491 | 0.377 | 0.598 | 0.519 | 0.491 | 0.354 | 0.519 | 0.519 |
| GPT-4o-Mini | 0.558 | 0.352 | 0.309 | 0.273 | 0.260 | 0.365 | 0.403 | 0.354 | 0.258 | 0.481 | 0.398 |
| GPT-3.5-Turbo-1106 | 0.567 | 0.332 | 0.325 | 0.259 | 0.292 | 0.422 | 0.424 | 0.352 | 0.294 | 0.408 | 0.403 |
| Grok-3 | 0.655 | 0.494 | 0.499 | 0.392 | 0.377 | 0.534 | 0.502 | 0.496 | 0.384 | 0.430 | 0.516 |
| Grok-3-Mini | 0.598 | 0.487 | 0.463 | 0.443 | 0.304 | 0.440 | 0.472 | 0.381 | 0.346 | 0.459 | 0.489 |
| Claude-Sonnet-4-5-20250929 | 0.708 | 0.552 | 0.576 | 0.524 | 0.463 | 0.622 | 0.597 | 0.565 | 0.476 | 0.598 | 0.596 |
| Claude-3-7-Sonnet-20250219 | 0.636 | 0.486 | 0.527 | 0.419 | 0.398 | 0.568 | 0.507 | 0.516 | 0.400 | 0.483 | 0.529 |
| Claude-3-5-Sonnet-20241022 | 0.599 | 0.457 | 0.462 | 0.509 | 0.393 | 0.549 | 0.498 | 0.454 | 0.391 | 0.526 | 0.520 |
| Claude-3-Haiku-20240307 | 0.563 | 0.374 | 0.373 | 0.479 | 0.357 | 0.345 | 0.419 | 0.396 | 0.337 | 0.472 | 0.445 |

*Table 16.* Efficiency results with Gemini-3-Pro-Preview. The most efficient result for each difficulty is bolded and underlined, while the least is highlighted in grey.

| Categorization | Time Efficiency ($\tau = 1.2, \sigma = 0$) | | | | Space Efficiency ($\tau = 0, \sigma = 1.2$) | | | | Time-Space Dual Efficiency ($\tau = \sigma = 1.2$) | | | |
|---|---|---|---|---|---|---|---|---|---|---|---|---|
| | Easy | Medium | Hard | Overall | Easy | Medium | Hard | Overall | Easy | Medium | Hard | Overall |
| String | **1.000** | 0.921 | 0.864 | 0.929 | **1.000** | 0.893 | 0.852 | 0.916 | **1.000** | 0.865 | 0.787 | 0.886 |
| Sort | **1.000** | 0.917 | 0.888 | 0.923 | **1.000** | 0.916 | 0.859 | 0.911 | **1.000** | 0.916 | 0.858 | 0.910 |
| Greedy | 0.987 | 0.894 | **0.976** | 0.951 | **1.000** | 0.877 | **0.950** | 0.935 | 0.941 | 0.880 | **0.944** | 0.922 |
| Graph | **1.000** | **0.964** | 0.840 | 0.900 | **1.000** | **0.996** | 0.812 | 0.896 | **1.000** | 0.938 | 0.730 | 0.830 |
| Data Structures | 0.858 | 0.900 | 0.843 | 0.863 | 0.875 | 0.821 | 0.771 | 0.800 | 0.818 | 0.853 | 0.760 | 0.797 |
| Two Pointers | 0.993 | 0.963 | 0.954 | **0.965** | **1.000** | 0.948 | 0.897 | **0.937** | 0.950 | **0.949** | 0.896 | **0.928** |
| Math | 0.904 | 0.906 | 0.867 | 0.888 | 0.911 | 0.906 | 0.805 | 0.862 | 0.868 | 0.901 | 0.784 | 0.841 |
| Search | **1.000** | 0.883 | 0.912 | 0.911 | **1.000** | 0.859 | 0.868 | 0.881 | **1.000** | 0.864 | 0.850 | 0.875 |
| DP | 0.977 | 0.927 | 0.824 | 0.868 | **1.000** | 0.924 | 0.793 | 0.849 | 0.924 | 0.907 | 0.757 | 0.816 |
| Other | **1.000** | 0.918 | 0.709 | 0.852 | **1.000** | 0.893 | 0.674 | 0.826 | **1.000** | 0.875 | 0.615 | 0.795 |
| Overall | 0.953 | 0.897 | 0.854 | 0.890 | 0.955 | 0.880 | 0.815 | 0.868 | 0.940 | 0.870 | 0.787 | 0.849 |

*Table 17.* Efficiency results with GPT-5.2. The most efficient result for each difficulty is bolded and underlined, while the least is highlighted in grey.

| Categorization | Time Efficiency ($\tau = 1.2, \sigma = 0$) | | | | Space Efficiency ($\tau = 0, \sigma = 1.2$) | | | | Time-Space Dual Efficiency ($\tau = \sigma = 1.2$) | | | |
|---|---|---|---|---|---|---|---|---|---|---|---|---|
| | Easy | Medium | Hard | Overall | Easy | Medium | Hard | Overall | Easy | Medium | Hard | Overall |
| String | 0.941 | 0.856 | 0.782 | 0.861 | 0.941 | 0.790 | 0.857 | 0.863 | 0.941 | 0.762 | 0.760 | 0.822 |
| Sort | **1.000** | 0.904 | 0.852 | 0.903 | **1.000** | 0.920 | 0.840 | 0.904 | **1.000** | 0.911 | 0.824 | 0.894 |
| Greedy | 0.957 | 0.844 | 0.910 | 0.897 | **1.000** | 0.844 | 0.868 | 0.883 | 0.940 | 0.844 | 0.867 | 0.872 |
| Graph | **1.000** | 0.964 | 0.696 | 0.821 | 0.999 | **1.000** | 0.697 | 0.834 | 0.994 | 0.930 | 0.614 | 0.763 |
| Data Structures | 0.875 | 0.900 | 0.785 | 0.833 | 0.875 | 0.864 | 0.768 | 0.812 | 0.839 | 0.837 | 0.712 | 0.768 |
| Two Pointers | 0.934 | 0.963 | **0.955** | **0.955** | **1.000** | 0.963 | **0.945** | **0.963** | 0.934 | **0.940** | **0.925** | **0.933** |
| Math | 0.882 | 0.906 | 0.826 | 0.865 | 0.913 | 0.906 | 0.778 | 0.851 | 0.869 | 0.891 | 0.742 | 0.819 |
| Search | **1.000** | 0.845 | 0.913 | 0.895 | **1.000** | 0.812 | 0.889 | 0.871 | **1.000** | 0.803 | 0.882 | 0.864 |
| DP | **1.000** | 0.896 | 0.778 | 0.831 | **1.000** | 0.885 | 0.739 | 0.804 | 0.952 | 0.868 | 0.678 | 0.758 |
| Other | **1.000** | **0.978** | 0.668 | 0.870 | **1.000** | 0.908 | 0.661 | 0.830 | **1.000** | 0.891 | 0.602 | 0.799 |
| Overall | 0.931 | 0.888 | 0.788 | 0.855 | 0.941 | 0.875 | 0.763 | 0.841 | 0.926 | 0.853 | 0.717 | 0.811 |

*Table 18.* Efficiency results with Gemini-2.5-Pro. The most efficient result for each difficulty is bolded and underlined, while the least is highlighted in grey.

| Categorization | Time Efficiency ($\tau = 1.2, \sigma = 0$) | | | | Space Efficiency ($\tau = 0, \sigma = 1.2$) | | | | Time-Space Dual Efficiency ($\tau = \sigma = 1.2$) | | | |
|---|---|---|---|---|---|---|---|---|---|---|---|---|
| | Easy | Medium | Hard | Overall | Easy | Medium | Hard | Overall | Easy | Medium | Hard | Overall |
| String | **1.000** | 0.941 | 0.783 | 0.911 | 0.957 | 0.898 | 0.731 | 0.865 | 0.957 | 0.890 | 0.658 | 0.839 |
| Sort | **1.000** | 0.958 | 0.731 | 0.876 | 0.968 | 0.946 | 0.715 | 0.858 | 0.968 | 0.943 | 0.713 | 0.856 |
| Greedy | 0.955 | 0.874 | 0.864 | 0.883 | **1.000** | 0.864 | 0.793 | 0.853 | 0.914 | 0.860 | 0.788 | 0.834 |
| Graph | **1.000** | **0.964** | 0.617 | 0.782 | **1.000** | **0.970** | 0.674 | 0.815 | **1.000** | 0.885 | 0.549 | 0.717 |
| Data Structures | 0.869 | 0.892 | 0.651 | 0.756 | 0.862 | 0.822 | 0.566 | 0.687 | 0.770 | 0.831 | 0.538 | 0.662 |
| Two Pointers | 0.934 | 0.963 | **0.880** | **0.926** | 0.934 | 0.962 | **0.850** | **0.914** | 0.868 | **0.961** | **0.828** | **0.893** |
| Math | 0.879 | 0.906 | 0.718 | 0.819 | 0.880 | 0.894 | 0.644 | 0.784 | 0.830 | 0.882 | 0.608 | 0.752 |
| Search | **1.000** | 0.893 | 0.717 | 0.830 | **1.000** | 0.893 | 0.633 | 0.794 | **1.000** | 0.876 | 0.648 | 0.793 |
| DP | 0.993 | 0.894 | 0.679 | 0.772 | **1.000** | 0.909 | 0.610 | 0.735 | 0.917 | 0.883 | 0.556 | 0.688 |
| Other | **1.000** | 0.941 | 0.562 | 0.812 | 0.999 | 0.898 | 0.560 | 0.788 | 0.999 | 0.880 | 0.541 | 0.771 |
| Overall | 0.944 | 0.900 | 0.693 | 0.825 | 0.926 | 0.883 | 0.648 | 0.796 | 0.904 | 0.867 | 0.597 | 0.764 |

*Table 19.* Efficiency results with DeepSeek-V3.2-Thinking. The most efficient result for each difficulty is bolded and underlined, while the least is highlighted in grey.

| Categorization | Time Efficiency ($\tau = 1.2, \sigma = 0$) | | | | Space Efficiency ($\tau = 0, \sigma = 1.2$) | | | | Time-Space Dual Efficiency ($\tau = \sigma = 1.2$) | | | |
|---|---|---|---|---|---|---|---|---|---|---|---|---|
| | Easy | Medium | Hard | Overall | Easy | Medium | Hard | Overall | Easy | Medium | Hard | Overall |
| String | **1.000** | 0.919 | 0.640 | 0.857 | 0.956 | 0.874 | 0.637 | 0.826 | 0.937 | 0.808 | 0.528 | 0.762 |
| Sort | **1.000** | 0.904 | 0.770 | 0.870 | **1.000** | 0.919 | 0.640 | 0.822 | 0.934 | 0.798 | 0.511 | 0.710 |
| Greedy | 0.992 | 0.866 | 0.820 | 0.865 | 0.999 | 0.846 | 0.717 | 0.809 | 0.931 | 0.776 | 0.623 | 0.727 |
| Graph | **1.000** | **0.963** | 0.647 | 0.794 | **1.000** | **0.977** | 0.650 | 0.800 | 0.982 | **0.879** | 0.468 | 0.664 |
| Data Structures | 0.857 | 0.801 | 0.554 | 0.671 | 0.874 | 0.741 | 0.547 | 0.650 | 0.788 | 0.645 | 0.377 | 0.514 |
| Two Pointers | **1.000** | 0.938 | **0.896** | **0.933** | 0.999 | 0.911 | **0.737** | **0.859** | 0.981 | 0.862 | **0.669** | **0.808** |
| Math | 0.907 | 0.864 | 0.683 | 0.793 | 0.912 | 0.875 | 0.583 | 0.754 | 0.883 | 0.785 | 0.497 | 0.680 |
| Search | **1.000** | 0.827 | 0.676 | 0.781 | **1.000** | 0.836 | 0.565 | 0.736 | **1.000** | 0.761 | 0.497 | 0.674 |
| DP | 0.975 | 0.881 | 0.664 | 0.755 | 0.998 | 0.902 | 0.589 | 0.717 | 0.885 | 0.824 | 0.468 | 0.610 |
| Other | **1.000** | 0.933 | 0.621 | 0.829 | **1.000** | 0.900 | 0.658 | 0.824 | 0.947 | 0.825 | 0.524 | 0.730 |
| Overall | 0.954 | 0.860 | 0.658 | 0.795 | 0.945 | 0.850 | 0.608 | 0.768 | 0.914 | 0.769 | 0.481 | 0.678 |

*Table 20.* Efficiency results with DeepSeek-R1. The most efficient result for each difficulty is bolded and underlined, while the least is highlighted in grey.

| Categorization | Time Efficiency ($\tau = 1.2, \sigma = 0$) | | | | Space Efficiency ($\tau = 0, \sigma = 1.2$) | | | | Time-Space Dual Efficiency ($\tau = \sigma = 1.2$) | | | |
|---|---|---|---|---|---|---|---|---|---|---|---|---|
| | Easy | Medium | Hard | Overall | Easy | Medium | Hard | Overall | Easy | Medium | Hard | Overall |
| String | **1.000** | 0.878 | 0.485 | 0.794 | **1.000** | 0.839 | 0.488 | 0.781 | 0.990 | 0.751 | 0.382 | 0.714 |
| Sort | **1.000** | 0.900 | 0.600 | 0.797 | **1.000** | 0.910 | 0.553 | 0.781 | 0.924 | 0.775 | 0.400 | 0.651 |
| Greedy | **1.000** | 0.800 | 0.711 | 0.791 | **1.000** | 0.765 | **0.626** | 0.737 | 0.943 | 0.689 | 0.518 | 0.648 |
| Graph | **1.000** | 0.960 | 0.520 | 0.715 | 0.985 | 0.914 | 0.549 | 0.714 | 0.973 | 0.819 | 0.423 | 0.610 |
| Data Structures | 0.875 | 0.790 | 0.335 | 0.548 | 0.875 | 0.727 | 0.286 | 0.501 | 0.839 | 0.619 | 0.228 | 0.430 |
| Two Pointers | **1.000** | 0.923 | **0.765** | **0.875** | **1.000** | **0.916** | 0.618 | **0.815** | 0.922 | 0.825 | **0.524** | **0.726** |
| Math | 0.913 | 0.890 | 0.498 | 0.721 | 0.913 | 0.808 | 0.474 | 0.684 | 0.880 | 0.733 | 0.406 | 0.622 |
| Search | **1.000** | 0.819 | 0.663 | 0.771 | **1.000** | 0.834 | 0.544 | 0.723 | **1.000** | 0.726 | 0.480 | 0.650 |
| DP | **1.000** | 0.877 | 0.480 | 0.639 | **1.000** | 0.894 | 0.453 | 0.628 | 0.949 | 0.829 | 0.367 | 0.550 |
| Other | **1.000** | **0.980** | 0.536 | 0.819 | **1.000** | **0.916** | 0.460 | 0.758 | 0.948 | **0.844** | 0.396 | 0.690 |
| Overall | 0.956 | 0.854 | 0.519 | 0.734 | 0.955 | 0.823 | 0.483 | 0.707 | 0.927 | 0.747 | 0.394 | 0.635 |

*Table 21.* Efficiency results with Gemini-2.5-Flash. The most efficient result for each difficulty is bolded and underlined, while the least is highlighted in grey.

| Categorization | Time Efficiency ($\tau = 1.2, \sigma = 0$) | | | | Space Efficiency ($\tau = 0, \sigma = 1.2$) | | | | Time-Space Dual Efficiency ($\tau = \sigma = 1.2$) | | | |
|---|---|---|---|---|---|---|---|---|---|---|---|---|
| | Easy | Medium | Hard | Overall | Easy | Medium | Hard | Overall | Easy | Medium | Hard | Overall |
| String | **1.000** | 0.933 | 0.758 | 0.900 | 0.674 | 0.842 | 0.681 | 0.733 | 0.674 | 0.817 | 0.603 | 0.700 |
| Sort | **1.000** | **0.960** | 0.748 | 0.882 | 0.717 | 0.866 | 0.601 | 0.726 | 0.717 | 0.868 | 0.562 | 0.711 |
| Greedy | 0.974 | 0.843 | 0.804 | 0.847 | 0.814 | 0.724 | 0.682 | 0.719 | 0.752 | 0.706 | 0.646 | 0.684 |
| Graph | **1.000** | 0.932 | 0.461 | 0.680 | 0.604 | 0.814 | 0.501 | 0.622 | 0.604 | 0.755 | 0.397 | 0.544 |
| Data Structures | 0.820 | 0.837 | 0.505 | 0.650 | 0.677 | 0.744 | 0.463 | 0.579 | 0.559 | 0.695 | 0.427 | 0.529 |
| Two Pointers | **1.000** | 0.952 | **0.884** | **0.934** | 0.784 | **0.919** | **0.704** | **0.812** | 0.774 | **0.888** | **0.677** | **0.786** |
| Math | 0.852 | 0.894 | 0.636 | 0.770 | 0.692 | 0.854 | 0.539 | 0.676 | 0.655 | 0.840 | 0.508 | 0.649 |
| Search | **1.000** | 0.880 | 0.566 | 0.755 | 0.802 | 0.841 | 0.438 | 0.656 | **0.802** | 0.817 | 0.430 | 0.642 |
| DP | 0.934 | 0.884 | 0.559 | 0.688 | **0.868** | 0.898 | 0.487 | 0.643 | 0.760 | 0.871 | 0.418 | 0.584 |
| Other | 0.667 | 0.935 | 0.418 | 0.726 | 0.535 | 0.816 | 0.434 | 0.653 | 0.535 | 0.798 | 0.382 | 0.625 |
| Overall | 0.935 | 0.878 | 0.608 | 0.777 | 0.695 | 0.805 | 0.536 | 0.671 | 0.678 | 0.780 | 0.477 | 0.633 |

*Table 22.* Efficiency results with DeepSeek-V3.2. The most efficient result for each difficulty is bolded and underlined, while the least is highlighted in grey.

| Categorization | Time Efficiency ($\tau = 1.2, \sigma = 0$) | | | | Space Efficiency ($\tau = 0, \sigma = 1.2$) | | | | Time-Space Dual Efficiency ($\tau = \sigma = 1.2$) | | | |
|---|---|---|---|---|---|---|---|---|---|---|---|---|
| | Easy | Medium | Hard | Overall | Easy | Medium | Hard | Overall | Easy | Medium | Hard | Overall |
| String | 0.941 | 0.882 | 0.506 | 0.782 | 0.933 | 0.882 | 0.560 | 0.796 | 0.917 | 0.800 | 0.425 | 0.720 |
| Sort | **1.000** | 0.904 | 0.643 | 0.818 | 0.999 | 0.913 | 0.523 | 0.772 | 0.921 | 0.788 | 0.437 | 0.673 |
| Greedy | 0.969 | 0.871 | 0.720 | 0.814 | 0.999 | 0.825 | 0.596 | 0.742 | 0.900 | 0.727 | 0.520 | 0.655 |
| Graph | **1.000** | **0.960** | 0.559 | 0.744 | **1.000** | **1.000** | 0.575 | 0.767 | 0.852 | **0.913** | 0.445 | 0.651 |
| Data Structures | 0.759 | 0.816 | 0.433 | 0.596 | 0.875 | 0.760 | 0.453 | 0.604 | 0.676 | 0.632 | 0.317 | 0.463 |
| Two Pointers | 0.909 | 0.952 | **0.776** | **0.876** | 0.908 | 0.928 | 0.638 | **0.812** | 0.875 | 0.859 | **0.577** | **0.753** |
| Math | 0.901 | 0.866 | 0.561 | 0.739 | 0.913 | 0.870 | 0.505 | 0.718 | 0.871 | 0.778 | 0.427 | 0.643 |
| Search | **1.000** | 0.873 | 0.716 | 0.819 | **1.000** | 0.861 | 0.596 | 0.761 | **0.952** | 0.772 | 0.537 | 0.690 |
| DP | 0.911 | 0.911 | 0.490 | 0.652 | **1.000** | 0.883 | 0.454 | 0.628 | 0.856 | 0.800 | 0.352 | 0.529 |
| Other | **1.000** | 0.813 | 0.617 | 0.761 | **1.000** | 0.816 | **0.652** | 0.776 | 0.904 | 0.745 | 0.516 | 0.679 |
| Overall | 0.920 | 0.849 | 0.543 | 0.736 | 0.939 | 0.837 | 0.501 | 0.718 | 0.878 | 0.747 | 0.404 | 0.630 |

*Table 23.* Efficiency results with O4-Mini. The most efficient result for each difficulty is bolded and underlined, while the least is highlighted in grey.

| Categorization | Time Efficiency ($\tau = 1.2, \sigma = 0$) | | | | Space Efficiency ($\tau = 0, \sigma = 1.2$) | | | | Time-Space Dual Efficiency ($\tau = \sigma = 1.2$) | | | |
|---|---|---|---|---|---|---|---|---|---|---|---|---|
| | Easy | Medium | Hard | Overall | Easy | Medium | Hard | Overall | Easy | Medium | Hard | Overall |
| String | **1.000** | 0.939 | 0.633 | 0.862 | 0.999 | 0.862 | 0.502 | 0.794 | **0.998** | 0.808 | 0.395 | **0.740** |
| Sort | **1.000** | 0.929 | 0.664 | 0.836 | 0.969 | 0.843 | 0.453 | 0.710 | 0.919 | 0.732 | 0.395 | 0.634 |
| Greedy | 0.997 | 0.869 | 0.764 | 0.840 | 0.988 | 0.808 | 0.511 | 0.693 | 0.915 | 0.744 | 0.455 | 0.631 |
| Graph | **1.000** | **0.961** | 0.490 | 0.706 | **1.000** | **0.960** | 0.443 | 0.680 | 0.937 | 0.790 | 0.337 | 0.556 |
| Data Structures | 0.870 | 0.876 | 0.467 | 0.648 | 0.875 | 0.804 | 0.291 | 0.528 | 0.800 | 0.684 | 0.266 | 0.466 |
| Two Pointers | 0.999 | 0.952 | **0.806** | **0.904** | 0.976 | 0.926 | **0.579** | **0.801** | 0.912 | **0.839** | **0.494** | 0.718 |
| Math | 0.908 | 0.899 | 0.550 | 0.746 | 0.892 | 0.834 | 0.481 | 0.691 | 0.866 | 0.803 | 0.412 | 0.644 |
| Search | **1.000** | 0.898 | 0.679 | 0.813 | **1.000** | 0.826 | 0.511 | 0.708 | **0.998** | 0.750 | 0.433 | 0.640 |
| DP | 0.977 | 0.880 | 0.515 | 0.663 | 0.998 | 0.813 | 0.378 | 0.560 | 0.896 | 0.786 | 0.300 | 0.496 |
| Other | 0.964 | 0.915 | 0.578 | 0.800 | 0.906 | 0.860 | 0.542 | 0.751 | 0.854 | 0.790 | 0.450 | 0.676 |
| Overall | 0.951 | 0.887 | 0.564 | 0.765 | 0.945 | 0.829 | 0.443 | 0.692 | 0.911 | 0.755 | 0.375 | 0.628 |

*Table 24.* Efficiency results with QwQ-32B. The most efficient result for each difficulty is bolded and underlined, while the least is highlighted in grey.

| Categorization | Time Efficiency ($\tau = 1.2, \sigma = 0$) | | | | Space Efficiency ($\tau = 0, \sigma = 1.2$) | | | | Time-Space Dual Efficiency ($\tau = \sigma = 1.2$) | | | |
|---|---|---|---|---|---|---|---|---|---|---|---|---|
| | Easy | Medium | Hard | Overall | Easy | Medium | Hard | Overall | Easy | Medium | Hard | Overall |
| String | 0.976 | 0.823 | 0.437 | 0.751 | 0.974 | 0.793 | 0.420 | 0.735 | 0.952 | 0.751 | 0.311 | 0.678 |
| Sort | **1.000** | 0.860 | 0.657 | 0.806 | 0.999 | 0.845 | 0.515 | 0.743 | 0.957 | 0.781 | 0.409 | 0.666 |
| Greedy | 0.971 | 0.794 | 0.714 | 0.785 | 0.973 | 0.731 | **0.582** | 0.700 | 0.895 | 0.704 | **0.506** | 0.639 |
| Graph | **1.000** | 0.873 | 0.538 | 0.702 | 0.999 | 0.830 | 0.539 | 0.687 | 0.982 | 0.694 | 0.397 | 0.559 |
| Data Structures | 0.875 | 0.798 | 0.449 | 0.613 | 0.875 | 0.735 | 0.331 | 0.529 | 0.814 | 0.594 | 0.274 | 0.444 |
| Two Pointers | 0.993 | 0.900 | **0.741** | **0.855** | 0.999 | 0.856 | 0.524 | **0.753** | 0.944 | 0.790 | 0.478 | **0.697** |
| Math | 0.892 | 0.833 | 0.550 | 0.720 | 0.894 | 0.779 | 0.449 | 0.658 | 0.848 | 0.732 | 0.396 | 0.608 |
| Search | **1.000** | 0.835 | 0.652 | 0.774 | **1.000** | 0.821 | 0.505 | 0.702 | **0.994** | 0.738 | 0.460 | 0.646 |
| DP | 0.931 | 0.890 | 0.481 | 0.642 | 0.927 | **0.905** | 0.394 | 0.592 | 0.839 | **0.874** | 0.344 | 0.545 |
| Other | **1.000** | **0.915** | 0.545 | 0.792 | **1.000** | 0.830 | 0.475 | 0.720 | 0.933 | 0.792 | 0.448 | 0.684 |
| Overall | 0.949 | 0.821 | 0.525 | 0.723 | 0.949 | 0.787 | 0.450 | 0.679 | 0.921 | 0.729 | 0.378 | 0.621 |

*Table 25.* Efficiency results with Claude-Sonnet-4-5-20250929. The most efficient result for each difficulty is bolded and underlined, while the least is highlighted in grey.

| Categorization | Time Efficiency $(\tau = 1.2, \sigma = 0)$ | | | | Space Efficiency $(\tau = 0, \sigma = 1.2)$ | | | | Time-Space Dual Efficiency $(\tau = \sigma = 1.2)$ | | | |
|---|---|---|---|---|---|---|---|---|---|---|---|---|
| | Easy | Medium | Hard | Overall | Easy | Medium | Hard | Overall | Easy | Medium | Hard | Overall |
| String | 0.918 | **0.973** | 0.471 | **0.794** | 0.941 | 0.907 | **0.559** | **0.807** | 0.895 | **0.852** | 0.358 | **0.708** |
| Sort | **1.000** | 0.820 | 0.421 | 0.695 | 0.996 | 0.811 | 0.384 | 0.676 | 0.942 | 0.636 | 0.272 | 0.552 |
| Greedy | 0.911 | 0.871 | 0.482 | 0.685 | 0.988 | 0.787 | 0.513 | 0.686 | 0.865 | 0.734 | **0.368** | 0.576 |
| Graph | **1.000** | 0.873 | 0.383 | 0.617 | **1.000** | 0.909 | 0.374 | 0.624 | 0.947 | 0.776 | 0.287 | 0.524 |
| Data Structures | 0.951 | 0.878 | 0.233 | 0.529 | **1.000** | **0.997** | 0.302 | 0.611 | 0.896 | 0.784 | 0.181 | 0.463 |
| Two Pointers | 0.927 | 0.875 | 0.487 | 0.731 | 0.980 | 0.849 | 0.441 | 0.710 | 0.887 | 0.786 | 0.328 | 0.622 |
| Math | 0.951 | 0.841 | 0.419 | 0.679 | 0.994 | 0.804 | 0.452 | 0.692 | 0.909 | 0.719 | 0.345 | 0.597 |
| Search | **1.000** | 0.779 | **0.504** | 0.685 | **1.000** | 0.808 | 0.436 | 0.666 | **1.000** | 0.670 | 0.342 | 0.565 |
| DP | 0.934 | 0.817 | 0.355 | 0.542 | **1.000** | 0.832 | 0.395 | 0.576 | 0.894 | 0.788 | 0.268 | 0.476 |
| Other | **1.000** | 0.872 | 0.470 | 0.750 | **1.000** | 0.788 | 0.465 | 0.704 | 0.975 | 0.698 | 0.306 | 0.598 |
| Overall | 0.963 | 0.858 | 0.389 | 0.684 | 0.982 | 0.844 | 0.423 | 0.697 | 0.927 | 0.741 | 0.303 | 0.596 |

