# OpenReview forum: "BEST: Benchmarking Efficiency in Space and Time for LLM-Generated Code"
_ICML.cc/2026/Conference — ICML 2026 regular_

### Official Review · Reviewer_RD9B · 2026-02-26

**Soundness:** 4
**Presentation:** 3
**Significance:** 3
**Originality:** 3
**Overall Recommendation:** 5
**Confidence:** 4

**Summary:**

This paper proposes BEST, the first benchmark that simultaneously evaluates the time and memory consumption of LLM-generated code. It comprises 440 expert-constructed coding tasks with a fine-grained subtask-based evaluation scheme and Pareto optimal baselines. The authors introduce a novel dual-indicator metric, dual@k, and evaluate 47 LLMs, revealing that while LLMs struggle with time efficiency, their performance on space efficiency is even worse.

**Compliance With Llm Reviewing Policy:**

Affirmed.

**Final Justification:**

I recommend an acceptance

**Key Questions For Authors:**

- Why using Base model for DeepSeek-Coder series model? For code generation, I suppose Instruct models are more proper for the non-completion tasks.
- It takes great effort and time for me to understand the proposed metric: Dual@K, maybe due to that authors make a long paragraph for setting the scene and explaining the motivation.
- Authors use C++ to evaluate the LLM-generated code, which is a standard choice. I wonder how much effort it might take to generalize to other PLs such Python or Java.
- The subtask-based evaluation relies heavily on expert-crafted test cases and Pareto optimal baselines. Given the plan to release new benchmark versions regularly to prevent data leakage, how scalable is this manual construction process?
- Code efficiency sometimes differs based on compiler optimizations (e.g., `-O2` vs `-O3`). Were compiler flags standardized during execution, and how might different optimization levels impact the space and time results?
- The benchmark is heavily biased toward algorithmic and competitive programming problems (e.g., LeetCode, Codeforces). It lacks coverage of repository-level software engineering tasks or practical, system-level development where space efficiency (e.g., memory leaks, object allocation overhead) is equally critical. After all, in the era of agents, benchmarking LLMs in more complex environments is important. However, given the huge effort of this paper, I think it is not a critical issue.
- The dual@k metric relies heavily on the hyperparameters $\tau$ and $\sigma$ to construct the weight matrix. While the authors justify $\tau = 1.2$ through an ablation study matching a hardness metric from prior literature, the parameter selection feels slightly heuristic and makes the absolute value of the metric difficult to interpret physically.
- The authors claim dual@k is essentially device-independent because it relies on pass rates rather than raw execution time. However, the binary pass/fail outcomes strictly depend on the execution limits, which are calibrated to the specific Intel Xeon hardware used for the evaluation. This claim feels a little bit overstated.

**Limitations:**

Yes

**Strengths And Weaknesses:**

# Strength
- Extensive experiments
- Pleasing results demonstration
- Comprehensive consideration for data collection
- Expert-generated test cases


# Weakness
- Scalability concerns
- Domain Limitation
- Heuristic Metric Parameters

---

> ### Author Rebuttal · Authors · 2026-03-31
>
> Thank you for your kind review and constructive suggestions, which help polish our manuscript. We are encouraged by your positive remarks. We hope the following responses can further dispel your concerns, and we have added more details to the revision.
>
> > **1. Model selections (Q1)**
>
> **A:**
> We aimed to keep our model selection aligned with prior efficiency benchmarks, such as Mercury [1], so we followed their setup. We agree that *Instruct* variants can be more appropriate, and we have evaluated the corresponding *DeepSeek-Coder Instruct* models under the same setup, as shown in Table 1 at [link](http://bit.ly/4bH20s6). Compared with Base models of the same parameter scale, the Instruct variants almost achieve slightly higher dual@k scores. This finding may show ''Instruct models are more proper for the non-completion tasks''. We have added these additional details and results in the revision.
>
> [1] Du et al. *Mercury: A Code Efficiency Benchmark for Code Large Language Models*
>
> > **2. dual@k is difficult to understand (Q2)**
>
> **A:**
> We are sorry that our current presentation makes the metric harder to follow. We believe more explanations do help. We have added more details in the introduction to improve clarity.
>
> > **3. Generalizing to other PLs (Q3/W1)**
>
> **A:**
> BEST can be easily generalized to other languages, such as Python or Java, by
> 1. adapting the prompts and interfaces to that language, and
> 2. re-evaluating the time and memory limits using expert baselines written in that language.
>
> Extending BEST to Python is already part of our ongoing work. We have taken a preliminary experiment, and the results show that although the absolute scores differ across languages, the relative rankings remain identical across C++ and Python. Due to the space limits, please refer to our response to **Reviewer #enub (W1)** for more details.
>
> > **4. BEST's scalability (Q4)**
>
> **A:**
> We agree that the first version of BEST involved substantial manual expert effort to ensure benchmark rigor and reliability. That said, during the later stages of its construction, we gradually identified several promising directions toward a more scalable pipeline for future BEST releases, such as exploiting high-quality human submissions, using LLMs for candidate drafting, and introducing semi-automatic screening to reduce the expert review burden.
>
> > **5. Compiler optimization flags (Q5)**
>
> **A:**
> In the manuscript, all experiments are conducted under a standardized compiler setting, with no optimization flags enabled. We agree that this detail should have been stated more explicitly. To further address this concern, we have additionally run experiments under compiler optimizations -O2 and -O3 on the same hardware. As shown in Table 3 at [link](http://bit.ly/4bH20s6), the results of the TOP5 models are consistent in rankings. This suggests that while optimization levels can change absolute runtimes, they do not materially change the main conclusions of the benchmark after the evaluation constraints are calibrated consistently.
>
> > **6. Domain limitation (Q6/W2)**
>
> **A:**
> We sincerely appreciate your recognition of ''the huge effort of this paper''. We agree that repository-level software engineering and system-level development are important future evaluation settings, especially in the era of coding agents. We also share this expectation and view the current work as a rigorous first step that establishes a controlled foundation for these future extensions.
>
> > **7. Heuristic hyperparameters (Q7/W3)**
>
> **A:**
> Our view is that $\tau$ and $\sigma$ are intended to provide a simple and controllable weighting scheme over the subtask matrix, allowing evaluators to place different emphasis on time and space flexibly.
>
> The default choice of 1.2 is selected mainly so that the numerical range of dual@k is aligned with prior metric eff@k, making the values easier to compare with previous work. We have further evaluated the sensitivity of the metric to different choices of $\tau$ and $\sigma$. As listed in Table 2 at [link](http://bit.ly/4bH20s6), the results show that the relative model rankings remain stable. This suggests that, although the absolute metric values vary, our main empirical conclusions and model ordering are not driven by a hyperparameter choice.
>
> > **8. Device-independence claim is overstated (Q8)**
>
> **A:**
> We are sorry for the imprecise wording in the manuscript. We agree that the claim should be stated more carefully. More precisely, dual@k is *device-independent after per-environment recalibration*, rather than absolutely invariant to hardware changes without recalibration. We have revised the wording accordingly.
>
> To further support our claim, we have conducted an additional experiment under *AMD EPYC 9004* and found the results to be consistent with those obtained in the original environment. Due to space limits, please refer to our response to **Reviewer #fdeg (Q1/W1)** and Table 3 at [link](http://bit.ly/4bH20s6) for details.

---

> > ### Author Rebuttal · Reviewer_RD9B · 2026-04-03
> >
> > Thanks authors for addressing my concern, I have raised my score. In the future, I'd like to see the space & time benchmarking can be transferred to larger scale, i.e., repo level.

---

> > > ### Author Response · Authors · 2026-04-08
> > >
> > > We sincerely thank you for your thoughtful review and for recognizing that we have addressed your concerns. We also appreciate your suggestion to extend the space and time benchmarking to larger-scale settings, such as the repository level, which we view as an important direction for future work. We have incorporated the clarifications and additional discussions from the rebuttal into the revised manuscript.

---

### Official Review · Reviewer_enub · 2026-03-05

**Soundness:** 4
**Presentation:** 3
**Significance:** 3
**Originality:** 3
**Overall Recommendation:** 5
**Confidence:** 4

**Summary:**

This paper introduces BEST, a benchmark for evaluating the efficiency of LLM-generated code in both time and space. The benchmark contains 440 expert-crafted coding tasks collected from various online coding platforms and originally created problems. Inspired by the International Olympiad in Informatics, the authors propose a subtask-based evaluation scheme where each task is divided into multiple subtasks with different input scales and time/space limits. The authors evaluate 47 LLMs and find that LLMs are generally weak at generating efficient code, and their space efficiency is even worse than time efficiency.

**Compliance With Llm Reviewing Policy:**

Affirmed.

**Key Questions For Authors:**

See weakness, it would be better if the paper can address more whether lack of space efficiency is limited to c/c++ or it's a general problem to all languages

**Limitations:**

Yes

**Strengths And Weaknesses:**

Strengths:
(1) Soundness: The benchmark construction is very rigorous. The task selection follows a two-stage process, the baselines are Pareto optimal and verified through a multi-stage process.The test cases are carefully designed by experienced algorithm contest participants. This level of rigor is rare in benchmark papers.

(2) Significance: The paper addresses a real gap in the field. Existing code benchmarks mainly focus on time efficiency/function correctness. Space efficiency is overlooked but matters a lot in practice.

(3) Originality: Evaluation metrics e.g. pareto curve,  dual@k  is well designed

Weakness:
(1) Significance: The evaluation is limited to C++ only. While it's accepted as this is a first benchmark, it is unclear whether the findings generalize to other languages. At least some discussion or small-scale experiment on other languages would strengthen the paper.

---

> ### Author Rebuttal · Authors · 2026-03-31
>
> Thank you for your thoughtful feedback and for noting that our benchmark ''addresses a real gap in the field''. We also appreciate your constructive suggestions, which have helped polish our manuscript. Below are our responses.
>
> > **1. Generalization beyond C++ (W1 / Q1)**
>
> **A:**
> Thank you for this important suggestion. We agree that whether our findings generalize beyond C++ is a key question.
>
> Our current version uses C++ because it provides a particularly clean setting for efficiency evaluation: it offers explicit memory management and fewer automatic runtime optimizations, making differences in time and space efficiency more directly observable. This allows BEST to more cleanly isolate *algorithmic efficiency* from language-runtime effects.
>
> Extending BEST to Python is already part of our ongoing work. As this extension is still under development, we provide a preliminary experiment here. We have implemented a Python instantiation of BEST on a subset of 100 tasks, recalibrated the corresponding time and memory limits using Python baselines, and evaluated several representative models, including GPT-5.2, Gemini-2.5-Pro, and DeepSeek-V3.2-Thinking, on this subset in both C++ and Python.
>
>
> Table: Comparison of dual@10 scores for C++ and Python instantiations on a subset of BEST. The relative rankings of the models remain identical across languages.
>
> | Model | Time Efficiency (C++) | Time Efficiency (Python) | Space Efficiency (C++) | Space Efficiency (Python) | Time-Space Dual Efficiency (C++) | Time-Space Dual Efficiency (Python) |
> |---|---:|---:|---:|---:|---:|---:|
> | GPT-5.2 | 0.867 | 0.891 | 0.853 | 0.809 | 0.823 | 0.796 |
> | Gemini-2.5-Pro | 0.841 | 0.858 | 0.811 | 0.774 | 0.776 | 0.751 |
> | DeepSeek-V3.2-Thinking | 0.806 | 0.834 | 0.779 | 0.737 | 0.691 | 0.696 |
>
>
> The results are shown in the table. Although the absolute scores differ across languages, the relative rankings remain identical across C++ and Python for time efficiency, space efficiency, and time-space dual efficiency.
>
> More importantly, the main qualitative finding also persists across languages: for all three models, *space-efficiency scores remain lower than time-efficiency scores* in both C++ and Python. This provides initial evidence that the observed weakness in space-efficient code generation is not limited to C++, but reflects a broader challenge in efficient code generation.
>
> We have included the complete setup, methodology, and analysis in the manuscript and will release the Python version online when finalized.

---

> > ### Author Rebuttal · Reviewer_enub · 2026-03-31
> >
> > The rebuttal fully resolved my questioins

---

> > > ### Author Response · Authors · 2026-04-08
> > >
> > > We sincerely thank you for your review and positive feedback. We are pleased that our rebuttal has fully addressed your questions. We have incorporated the clarifications and additional discussions from the rebuttal into the revised manuscript.

---

### Official Review · Reviewer_ncxj · 2026-03-13

**Soundness:** 3
**Presentation:** 4
**Significance:** 3
**Originality:** 2
**Overall Recommendation:** 5
**Confidence:** 4

**Summary:**

The paper proposes a new rigorous eval benchmark called BEST to systematically evaluate LLM generated code's space and time complexity containing 440 problems with expert written solutions and subtask definition and cost constraints. Further, the paper differentiates itself from prior work on EffiBench by measuring "dual@k" - a generalization of the commonly used "pass@k" metric to evaluate LLM code's adherence to both space and time constraints and not just to either of them.

**Compliance With Llm Reviewing Policy:**

Affirmed.

**Final Justification:**

I recommend an acceptance.

**Key Questions For Authors:**

N/A

**Limitations:**

Yes

**Strengths And Weaknesses:**

Strengths:

1. The comprehensive experimental brevity of the paper must be applauded.
2. The motivation is clear and simple.
3. dual@k is a neat extension of pass@k for this setup.
4. The paper is well written and easy to follow.
5. Ablations cover a lot of the initial questions I had with the main results.

Weaknesses:

1. The device agnostic claim is a bit overstated - pass/ fail is still based on enforcing time/ memory limits which can be device/ OS/ hardware/ sandbox specific. I am not sure if it is possible to fully disentangle this as much as the paper claims it to be.
2. Measuring space efficiency via global memory limits is disadvantageous in my evaluation since it can conflate much higher memory usage caused to simple factors like package usage or container overheads for instance.
3. While the dual@k metric is a neat extension and stands out in terms of novelty, the same can't be stated about the benchmark BEST itself. I feel it would have been possible to simply show dual@k on Effibench? And hence in doing so, show also the benchmark correlation between Effibench and BEST - thus making one of them redundant? I don't think the subtask granularity and expert handcrafted solutions is enough a differentiating factor - although it is appreciated to have them.

---

> ### Author Rebuttal · Authors · 2026-03-31
>
> Thank you for your constructive reviews. We are encouraged by your appreciation of our efforts and by your comment that ''the comprehensive experimental brevity of the paper must be applauded''. Below, we clarify the concerns and have revised the manuscript accordingly.
>
> > **1. Device-agnostic claim (W1)**
>
> **A:**
> Thank you for this helpful comment, and we are sorry for the imprecise wording in the manuscript. We agree that the claim should be stated more carefully. More precisely, dual@k is *device-independent after per-environment recalibration*, rather than absolutely invariant to hardware changes without recalibration. We have revised the wording accordingly.
>
> The key point is that dual@k is not computed from raw runtime or raw memory values. Instead, it aggregates subtask-level pass/fail outcomes under benchmark-defined time/space thresholds. Thus, hardware and system differences mainly affect the calibration of those thresholds, not the definition of the metric itself.
>
> BEST also mitigates hardware/compiler bias through per-environment baseline recalibration and input-scale design; due to space limits, please refer to our response to **Reviewer #fdeg (Q1)** for details.
>
> To further support our claim, we have conducted an additional experiment under *AMD EPYC 9004* and found the results to be consistent with those obtained in the original environment. Due to space limits, details are also in our response to **Reviewer #fdeg (Q1)** and Table 3 in [link](http://bit.ly/4bH20s6).
>
> > **2. Space efficiency measurement (W2)**
>
> **A:**
> Thank you for this thoughtful concern. We agree that in general software settings, memory usage can indeed be confounded by package overhead, framework behavior, or container/runtime overheads. However, BEST is intentionally scoped to self-contained C++ algorithmic tasks precisely to minimize these confounds.
>
> In our setting, all submissions are evaluated under the same language, compiler, and sandbox, and the tasks are not package-heavy or framework-dependent. Therefore, fixed overhead is approximately shared across solutions, while the dominant differences come from algorithmic choices, such as in-place computation vs. auxiliary arrays, or brute force vs. data structures.
>
> This is exactly the notion of *space efficiency* that BEST aims to measure: whether the model can choose algorithms and data structures that achieve better time-space trade-offs.
>
> In addition, our subtask-specific memory caps are designed as *generous upper bounds*, so minor constant overheads are less likely to determine pass/fail, while substantial differences in asymptotic space usage are still separated. In other words, BEST is not intended to measure end-to-end systems memory in package-rich, multi-file projects; it is intended to isolate *algorithmic space efficiency* in a controlled setting. We have clarified this scope more explicitly in the paper.
>
> > **3. Necessity of BEST beyond dual@k (W3)**
>
> **A:**
> Thank you for recognizing the novelty of dual@k. We agree that this point deserves clearer clarification, and we have added the following discussion to the related work section of the manuscript.
>
> While one could apply a modified scoring rule to an existing benchmark such as EffiBench, this would still **not** capture the main goal of our work. dual@k is designed for a benchmark with a *two-dimensional subtask structure* and explicit *time-space trade-offs*, which prior benchmarks generally do not provide. BEST is built precisely to supply this structure.
>
> BEST differs from existing benchmarks in several essential ways:
> 1. BEST evaluates time-space trade-offs, not just time efficiency. Rather than rewarding only the fastest solution under a single dominant constraint, BEST focuses on tasks with multiple valid algorithms exhibiting different time/space trade-offs.
> 2. The subtask matrix is essential for dual@k. Since dual@k is defined over subtasks with different time and space ranks, without this structure, it would largely collapse into a one-dimensional efficiency score.
> 3. Multiple Pareto baselines are necessary for fair evaluation. Because no single implementation is optimal in both time and space, BEST uses expert-crafted Pareto-optimal baselines to avoid penalizing valid but different efficiency choices.
> 4. BEST is evaluation-oriented by design. Its tasks are selected or created to admit meaningful alternative implementations and to distinguish efficiency behaviors under controlled subtasks.
>
> Therefore, BEST provides the benchmark structure that makes dual@k meaningful for joint time-space evaluation. More broadly, it contributes a benchmark setting that goes beyond single-objective efficiency assessment and enables evaluation of resource-adaptive code generation across explicit time-space trade-offs.

---

> > ### Author Rebuttal · Reviewer_ncxj · 2026-04-03
> >
> > Thank you to the authors for the detailed response and supporting evidence for clarification. All my concerns have been resolved. I will raise my score accordingly.

---

> > > ### Author Response · Authors · 2026-04-08
> > >
> > > We sincerely thank you for your careful review and encouraging feedback. We are grateful that our detailed response and supporting evidence have resolved your concerns. We have incorporated the clarifications and additional discussions from the rebuttal into the revised manuscript.

---

### Official Review · Reviewer_fdeg · 2026-03-13

**Soundness:** 3
**Presentation:** 3
**Significance:** 4
**Originality:** 3
**Overall Recommendation:** 5
**Confidence:** 3

**Summary:**

The paper focuses on a gap in the evaluation of code-generating LLMs where previous LLM efficiency benchmarks mostly emphasize time efficiency but largely overlook space efficiency. It introduces BEST, a 440-task benchmark with varying time and space constraints. The paper also proposes dual@k, a metric that aggregates subtask-level pass@k scores. The empirical study covers 47 LLMs and finds that current models are less capable of generating space-efficient code.

**Compliance With Llm Reviewing Policy:**

Affirmed.

**Final Justification:**

I recommend an acceptance. The additional clarification and cross-environment experiment adequately resolved my concern.

**Key Questions For Authors:**

1. Do you have cross-hardware or cross-compiler results that would further support the claim that the benchmark is essentially device-independent?
2. Since many tasks are sourced from public platforms, did the authors conduct a contamination analysis to mitigate the concern?

**Limitations:**

yes

**Strengths And Weaknesses:**

Strengths:
1. The paper is clearly organized and easy to follow. The appendix provides substantial details.
2. The benchmark is carefully designed and comprehensive. There are expert-created tasks with expert-crafted Pareto-optimal baselines. This fills a meaningful gap in LLM code-efficiency evaluation.
3. The empirical analysis is extensive, covering 47 models. The large-scale evaluation provides evidence that almost all tested models perform worse on space efficiency than on time efficiency.

Weaknesses:
1. It is unclear whether the benchmark is truly device-independent. Extra experiments across different hardware platforms and compilers would help support this claim.
2. Tasks are selected from various online platforms plus expert-created tasks. This raises concerns about potential data contamination.

---

> ### Author Rebuttal · Authors · 2026-03-31
>
> Thank you for your thoughtful review and for recognizing that BEST ''is carefully designed and comprehensive'' and ''fills a meaningful gap in LLM code-efficiency evaluation''. We are encouraged by these positive remarks. Below, we respond to your concerns and have incorporated these clarifications into the revision.
>
> > **1. Device-independence (W1/Q1)**
>
> **A:**
> Thank you for raising this important point. To validate our device-independent claim, we conduct a preliminary experiment and provide additional empirical results and clarifications below. We hope these results can help address the concern.
>
> We run a cross-environment check on the full benchmark using *AMD EPYC 9004*, as Table 3 at [link](http://bit.ly/4bH20s6). After re-timing the expert baselines and recalibrating the limits on the environment, the resulting model rankings remain consistent. We have added these results to the revision. These results further support our claim that BEST is largely hardware/compiler agnostic at the level of model ranking and comparative evaluation.
>
> We further note that dual@k is device-independent precisely because it is defined not on raw runtime or memory values, but on subtask-level pass/fail outcomes under benchmark-specified time and space thresholds. Consequently, hardware and system differences mainly influence the calibration of these thresholds, rather than the metric definition itself.
>
> More concretely, BEST reduces hardware/compiler bias in two ways:
>
> 1. Relative limits from expert baselines. For each subtask, we run the expert near-optimal baseline on the target environment and set the time/space limits as a generous multiple of that run (e.g., around 10x). If the benchmark is ported to another machine or compiler, the baselines can be re-timed, and the limits recalibrated accordingly.
> 2. Separation by asymptotic complexity. The input scales are chosen to separate complexity classes. An asymptotically slower algorithm may benefit from a smaller constant factor, but at the selected scales, it should still fail stricter subtasks, whereas an algorithm with the correct complexity should pass robustly. Therefore, the evaluation is designed to depend mainly on algorithmic efficiency, rather than small machine-specific constant-factor differences.
>
>
> > **2. Data contamination (W2/Q2)**
>
> **A:**
> We agree that contamination is an important concern for any benchmark built partly from public platforms, and we have taken action to mitigate this risk.
>
> Our mitigation has three parts:
> 1. Mixed-source construction. BEST is not a direct scrape from a single public platform; it combines carefully selected tasks from multiple online judges with *expert-created tasks*.
> 2. Benchmark-specific evaluation targets. Even when a problem statement is public, the evaluation target in BEST is not merely to reproduce a single canonical online solution. Each task is expanded into multiple subtasks with benchmark-specific input scales, time/space limits, expert-crafted Pareto-optimal baselines, and targeted test cases. As a result, memorizing one public solution does not trivially translate into a high dual@k score, especially on harder subtasks that probe time-space trade-offs.
> 3. Benchmark refresh. As stated in the paper, we plan to release refreshed versions of BEST periodically to reduce long-term leakage risk further.
>
> To further assess this issue, we conducted a split analysis on
>  - expert-crafted tasks (assuming that they do not leak).
>  - newly selected tasks (from the latest Codeforces problems).
>
> We present the means and variances of dual@k for three representative LLMs at different levels of difficulty in Figure 1 at [link](http://bit.ly/4bH20s6). The performance trends are consistent with those on the full benchmark.
>
> Therefore, while no benchmark with public-source components can fully rule out contamination, we believe contamination is unlikely to explain the main findings of BEST.

---

> > ### Author Rebuttal · Reviewer_fdeg · 2026-04-04
> >
> > The rebuttal provided detailed clarification and supporting materials. This has resolved my concerns for the device-independence claim and data contamination. I really appreciate the split analysis for different categories which addressed the concern for potential data contamination.

---

> > > ### Author Response · Authors · 2026-04-08
> > >
> > > We sincerely thank you for your thoughtful review and constructive feedback. We are glad that our rebuttal and additional split analysis have resolved your concerns about device-independence and potential data contamination. We have incorporated the clarifications and additional discussions from the rebuttal into the revised manuscript.

---

### Decision · Program_Chairs · 2026-04-30

**Decision:**

Accept (regular)

**Comment:**

This paper presents BEST, a new benchmark comprising of 440 coding tasks to evaluate code generation with LLMs that evaluates for both time and space. It also proposes a fine-grained evaluation scheme where each task is divided into multiple sub-tasks with different input scales and complexity, and each of the sub-task is accompanied by an expert implementation that achieves pareto optimality. Finally, the paper evaluates 47 LLMs on BEST using a novel dual@k metric that generalizes pass@k metric and takes into account both time and space that captures the code efficiency in both space and time. All reviewers appreciated the key contributions of the paper of a new benchmark that is going to be refreshed regularly and comprehensive set of experiments to evaluate different LLMs. There were a few questions around hardware-specific nature of the tasks, data contamination, generalization to languages like Python beyond just C++, motivation for a new benchmark etc., but the rebuttal response helped alleviate most of these concerns. It would be great to add the new experiments and additional explanations in the final version of the paper.